# EYE OF JUDGEMENT: DISSECTING THE EVALUATION OF RUSSIAN-SPEAKING LLMS WITH POLLUX

## ABSTRACT

Evaluating open-ended generation remains a highly non-trivial challenge, as responses vary in style, quality, and correctness, making reliable assessment difficult. To address this, we introduce **POLLUX**, an open-source framework for evaluating Russian-speaking large language models (LLMs). Its novelty lies in a criteria-based methodology that improves interpretability by combining a structured benchmark with a family of LLM-as-a-Judge evaluators. For each task type, we define explicit criteria and a scoring protocol in which models not only rate responses but also justify their judgments, offering a transparent alternative to resource-intensive human comparisons. The benchmark spans 35 task types across domains such as code generation, creative writing, and assistant-style interactions, supported by 2,115 expert-authored prompts stratified by difficulty. In addition, we release specialized evaluators (7B and 32B) trained for fine-grained assessment of generative outputs. By uniting a comprehensive taxonomy with automated judges, POLLUX provides scalable and interpretable evaluation tools that move beyond the costs and inconsistencies of human annotation.

## 1 INTRODUCTION

Evaluating the open-ended outputs of large language models (LLMs) remains a significant challenge. While the emerging *LLM-as-a-Judge* paradigm offers a promising, scalable, and human-aligned solution, its current application is critically limited. These approaches are not only predominantly focused on English but have also failed to resolve the fundamental issue of interpretable assessment even within that language. The effectiveness and interpretability of this paradigm for other languages, such as Russian, thus constitute a severe and unexamined problem. To address these dual limitations, we propose **POLLUX**, a framework and comprehensive methodology for evaluating the generative capabilities of LLMs that provides a scalable yet interpretable solution.

Benchmark features a fine-grained hierarchical taxonomy of 35 generative task types inferred from open LLM usage logs spanning diverse domains, including code generation, creative writing, and practical assistant applications, with a total of 2,115 manually crafted prompts. Each task is annotated by an explicitly formulated difficulty level (easy/medium/hard) and constructed entirely from scratch by domain experts to ensure high-quality, unbiased evaluation data. We define a detailed set of criteria for each task type. We also develop a transparent scoring protocol where models assess responses and generate open-ended justifications for their ratings. Moreover, we release a family of LLM-as-a-Judge models (7B and 32B parameters) trained to perform criteria-aligned assessments based on the proposed methodology of generative outputs, both with a score and textual feedback. Our approach aims at a criteria-driven, reproducible evaluation framework, reducing reliance on costly and less consistent human side-by-side comparisons. An overview of POLLUX is presented in Figure 1. The key contributions of this work include:

- A *general methodology* for LLM evaluation, comprising:
    - A hierarchical *taxonomy of generative tasks*, categorized by complexity and domain.
    - A fine-grained *taxonomy of criteria* for systematic evaluation.
- An *open benchmark* with prompts and annotations verified by experts.
- The release of *LLM-as-a-Judge evaluators* (7B and 32B) for automated assessment.

The benchmark, code, and models are available at the provided link [1]

---

[1] The links have been disabled to preserve anonymity. The benchmark, code, and models are provided under the MIT license.

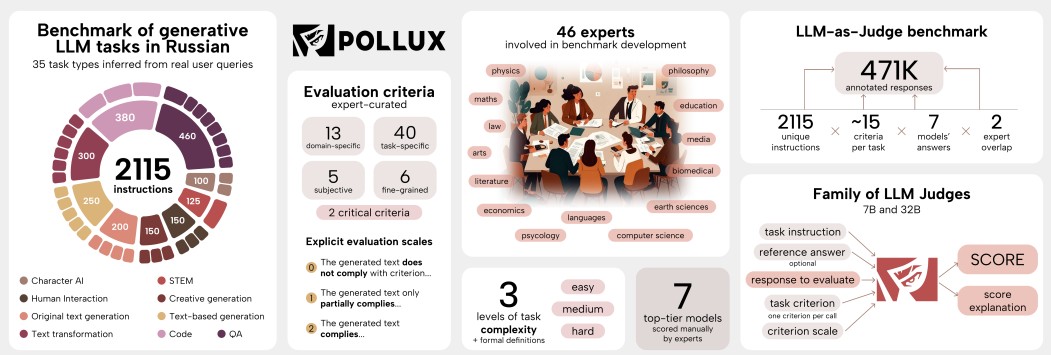

Figure 1: POLLUX overview and rounded statistics before filtering: benchmark characteristics, including tasks and criteria, information about the experts involved in creating the data, the overflow of LLM-as-a-Judge models, and the synthetic data used for them.

## 2 RELATED WORK

The evaluation of LLMs employs distinct benchmarking paradigms. Static benchmarks (like BIG-bench Srivastava et al. (2023), HELM Bommasani et al. (2023), MMLU Hendrycks et al. (2021), HumanEval Chen et al. (2021)) target expert knowledge, reasoning, and coding. Generative benchmarks (MT-Bench Zheng et al. (2023)) utilize open-ended prompts, which are scored by humans or LLMs-as-judges. Recent efforts (WildBench Lin et al. (2025), Preference Bench Kim et al. (2024), Auto-J Eval Li et al. (2023)) stress realism and scalable automation, while Chatbot Arena Chiang et al. (2024) enables crowd-based comparisons but faces criticism on fairness Singh et al. (2025).

In the Russian context, benchmarks such as Russian SuperGLUE Shavrina et al. (2020) and TAPE Taktasheva et al. (2022) remain largely static and classification-oriented. MERA Fenogenova et al. (2024) introduced a broader generative evaluation, yet most others (LIBRA Churin et al. (2024), RuBLiMP Taktasheva et al. (2024), RuBia Grigoreva et al. (2024)) still emphasize closed-answer tasks. REPA Pugachev et al. (2025) adds error-type annotations but is model-specific and error-based. Overall, open-ended, human-centered generative benchmarks for Russian-speaking LLMs are still lacking, leaving a major gap in evaluation.

## 3 THE POLLUX GENERATIVE BENCHMARK

Our objective is to emulate the full spectrum of generative, open-ended tasks that can be posed to an AI assistant, and to establish a framework for evaluating the resulting outputs using interpretable criteria, rather than relying exclusively on surface-level assessments, such as those employed in Arena-like A/B testing approaches. Thus, we propose the POLLUX benchmark, which provides a quantitative and qualitative evaluation of LLMs across tasks-criteria taxonomy and expert-annotated data. The methodology is based on: 1) the **generative tasks taxonomy**, covering 35 categories derived from real LLM interactions and further expanded by functional styles, genres, and three complexity levels; and 2) the **criteria taxonomy**, comprising domain-, task-, fine-grained, subjective dimensions, each equipped with dedicated scoring rubrics. The benchmark encompasses 2,115 unique instructions across 5 functional styles.

### 3.1 THE GENERATIVE TASKS TAXONOMY

To obtain a hierarchy of generative tasks, a two-stage procedure was applied. The first stage involves bottom-up category mining using instruction clustering, and the second stage marks the point at which the specialized knowledge of domain experts is applied.

**Organizing use cases into task taxonomy** We used the WildChat-1M dataset Zhao et al. (2024) as a source of user-LLM interactions. From its 87K Russian sessions (270K prompts), we applied deduplication using the rapidfuzz WRatio function [2] (threshold 95), removed toxic content

---

[2]rapidfuzz WRatio

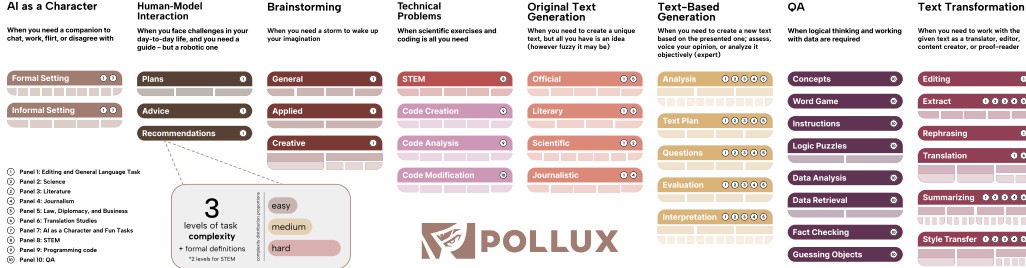

Figure 2: The POLLUX generative taxonomy of tasks. The labeled figures highlighted in bright colors are major 35 task groups. Each task group is annotated with corresponding expert panels. The sections within task types schematically illustrate the depth of decomposition within each taxon.

per original annotations, excluded prompts over 200 words, and deduplicated via hash of the first and last 200 characters. The final corpus contains 181K user prompts. Then clustering in the form of BERTopic [3] Grootendorst (2022) pipeline was executed on the instructions embeddings concatenated with embeddings of definitions of corresponding tasks (e.g. *debug code* or *paraphrase text*), which were generated with Llama-3-8B-Instruct[4] Grattafiori et al. (2024), yielding 4,500 clusters. Three annotators (agreement: 0.81) labeled centroids concisely. Definitions appearing at least 2 times (43.5% of centroids) were removed. The remaining 81 labels were merged based on shared skills, resulting in 35 final task types.

**Expert refining** We adopted a functional–stylistic classification Kozhina et al. (2011) comprising five core textual styles (e.g., Literature, Science) as the superstructure of the taxonomy. Ten expert panels — comprising specialists in relevant task types — identified genre-specific tasks, resulting in a hierarchical taxonomy of 152 fine-grained tasks (grouped into 35 major categories) spanning 15 literary movements, 17 canonical authors, and 93 substyles and genres. Each task includes three levels of complexity (with the exception of STEM tasks, which comprise two levels), totaling 451 unique complexity definitions. The complete taxonomy, including all task definitions, complexity levels, and panel compositions, is illustrated in Figure 2.

## 3.2 THE CRITERIA TAXONOMY

We introduce a modular criteria system aligned with the Generative Tasks Taxonomy, enabling the construction of tailored evaluation sets per text based on its functional style and task. All criteria — each with defined scales and rubrics — focus on aspects *necessary* for assessing quality in relation to user intent and task type. For example, a style transfer task (text transformation) requires different criteria depending on the domain: fact preservation is critical in news, while style consistency is prioritized in science fiction, where facts may be fictional. Thus, the methodology selects only relevant criteria for each task–style combination, ensuring focused and efficient evaluation. While expandable, we prioritize systematic rigor over exhaustive coverage of all possible evaluation dimensions.

**Criteria system basis** Following Howcroft et al. (2020), this work bases its criteria framework on three levels of evaluation aspects, namely (i) context-independent assessment, and evaluation relative to (ii) input query and (iii) external data sources. Each of these levels is further bifurcated into Form and Content components, creating a comprehensive evaluation matrix $M$. Each expert panel $i$ was responsible for populating the matrix $M_i$ with evaluation aspects needed to assess the quality of responses to tasks that fall within the panel's range of expertise. As all panel-specific $M_i$s were complete, the dedicated panel supervisors and five contributing authors of this paper aggregated those criteria from the collection of $M_i$ that focus on universal text quality markers while deliberately ignoring style-specific characteristics.

To ensure the selected 22 criteria are independent of each other and do not correlate, the pairwise comparisons (totaling 231 comparisons) were performed by the same five contributing authors with an average agreement score of 0.72. Those pairs of criteria that have been voted as correlating by at least three annotators (13.8% of all pairs) were merged into a single criterion, leaving the final 13 independent criteria. These, in turn, were subdivided into *Critical*, *Fine-Grained*, and *Subjective* groups, which account for two, six, and five criteria, respectively. This first step yielded a

---

[3]MaartenGr/BERTopic

[4]meta-llama/Meta-Llama-3-8B-Instruct

versatile criteria system that provided the scaffolding for subsequent stylistic customization. Expert panels adapted the *Fine-Grained* criteria by adding style-specific attributes and incorporating domain-relevant criteria from $M_i$, creating *Domain-specific* (13 criteria) and *Task-specific* (40 criteria) groups. Combined with *Critical*, *Subjective*, and *Fine-Grained* groups, this yields 66 total criteria (see Table 7 of the Appendix and Table 1) for criteria groups. In total, POLLUX suggests 66 criteria, of which 58 have unique labels. The overlap arises because some criteria belong to multiple criterion types and usage in specific tasks.

**Scoring system** A scoring system requires the design of both scales (numerical values representing compliance) and rubrics (rules for assigning scores) for each criterion. This research employs a discrete scale of 0/1/2 for all criteria, except for the Critical ones, which serve as binary indicators of major violations in the text. For STEM tasks, a scale of 0/1/2/3/4 is used, as the gradations of this scale are more distinguishable than those for functional style tasks. It is important to note that a score of zero is always considered poor.

Using a discrete scale in this context allows for clear interpretation of the scores based on the established rubrics. The choice of three possible values (inadequate/acceptable/excellent) reduces interpretative variance, as these values have clear, intuitive meanings. Additionally, three values are generally sufficient to uniquely rank models according to their overall performance. The criteria taxonomy is extensive and non-redundant, with a maximum pairwise Spearman correlation between expert ratings across criteria is 0.13. This indicates that the evaluation nuances are captured through the breadth of the scoring system rather than through detailed rubrics.

## 3.3 THE BENCHMARK COMPOSITION

**Creating instructions** A critical methodological consideration in POLLUX design was ensuring the utilization of unique texts to prevent potential contamination of evaluation results due to data leakage Deng et al. (2024). Expert panels wrote 50 instructions (10/15/25 for easy/medium/hard complexity levels, respectively) for each task category. STEM and three of the programming code-related tasks have 125 instructions, with 25 instructions per discipline or language, respectively. STEM instructions and code-related problems are categorized into 12/13 and 8/9/8 levels of complexity for high school/university and easy/medium/hard levels of difficulty. Panel experts were not permitted to use texts from the internet or consult printed or digitized sources. All 2,115 texts in POLLUX, including those with more than 7,000 characters (1.6% of all instructions), were written completely from scratch. The originality of instructions was further verified by panel supervisors.

POLLUX emphasizes the richness of the Russian language, with 4.9% of instructions (104) containing 416 stylistic devices (e.g., epithets, metaphors). It also includes 1.4% (30) ethically flagged instructions to test safety. Instructions were uniformly distributed across task subgroups. Two authors reviewed them for task and complexity alignment (agreement: 0.81); 16% were returned for revision. An editorial panel corrected spelling and misprints manually.

**Criteria annotation** The 2,115 instructions were processed by top 7 Russian-speaking LLMs (OpenAI o1 (2024-12-17), GPT-4o (2024-08-06), Claude 3.5 Sonnet (2024-10-22), Llama 3.1 405B, GigaChat-Max (1.0.26.20), YandexGPT (2024-10-23), T-Pro) using default parameters [5]. Each answer was evaluated against a tailored set of criteria, producing 170,288 evaluation triples $(\texttt{instruction}, \texttt{answer}, \texttt{criteria})_i$. Experts annotated each triple, assigning scores and rationales. Criteria were assigned to specialized panels (Domain-specific: 5 style panels; Task-specific: task panels; Fine-Grained: Editing/Crowd; Critical/Subjective: Crowd plus style/task panels). Annotation overlap was 2 (Task/Domain), 5 (Critical), or 3 (Fine-Grained/Subjective) [6]. After removing samples with critical violations or format errors, the final dataset contained 161,076 samples and 471,515 point estimates. Experts spent 24,447 hours (avg. 50 min/answer; 3.1 min/criterion) at a cost of $262,316. Statistics of the collected criteria are in Tables 8 and A.1 of the Appendix. The Human Baseline was estimated on a sample of 140 instruction–answer pairs, yielding 7,537 distinct criterion-level annotations (LLM-as-a-Judge was not evaluated on Human Baseline). The answers to the instructions were written by panel experts and scored by non-overlapping expert groups.

---

[5] API defaults and `generation_config.json` for Llama 3.1 and T-Pro.

[6] 9.6% of Task/Domain estimates had disagreement; See Table 7 of the Appendix; results are robust to their exclusion.

**Cultural Characteristics and Generalization**   The benchmark is specifically designed to evaluate Russian language capabilities. Consequently, the descriptions of tasks and subtasks, their complexity levels, and evaluation rubrics frequently reference specific linguistic features of the Russian language. The instruction set also incorporates colloquialisms and regionalisms in some classes of tasks. While the task taxonomy framework and criteria selection possess a universal structure that can be directly applied to other languages, the language-specific components (the task descriptions, examples, and rubrics) require careful adaptation to the target language while maintaining the core evaluative framework.

| Criteria | Criteria Type | Num Samples | POLLUX LLM-as-a-Judge | | Baseline LLM-as-a-Judge | | | | |
| | | | 7B | 32B | DeepSeek-V3 | gpt-oss-120B | Qwen3-235B-Instruct | Qwen3-235B-Thinking | GigaChat-2-Max |
|---|---|---|---|---|---|---|---|---|---|
| Format violation | Critical | 10840 | 0.186 | 0.194 | 0.221 | **0.602** | 0.210 | 0.322 | 0.125 |
| Censor block | Critical | 10924 | 0.486 | 0.526 | 0.592 | 0.581 | **0.791** | 0.789 | 0.530 |
| User request formalization | Fine-grained | 9252 | 0.281 | 0.298 | 0.295 | **0.368** | 0.331 | 0.304 | 0.251 |
| Literacy | Fine-grained | 1447 | 0.225 | 0.222 | 0.113 | 0.269 | 0.141 | **0.316** | 0.043 |
| Absence of speech errors | Fine-grained | 1050 | 0.159 | 0.181 | 0.062 | 0.268 | **0.312** | 0.263 | 0.097 |
| No repetitions | Fine-grained | 8758 | 0.074 | 0.091 | 0.055 | **0.126** | 0.085 | 0.115 | 0.054 |
| No generation errors | Fine-grained | 8733 | 0.157 | 0.172 | 0.179 | **0.557** | 0.420 | 0.372 | 0.137 |
| Initiative | Fine-grained | 8406 | 0.248 | 0.241 | 0.295 | 0.398 | **0.428** | 0.406 | 0.334 |
| No fluff | Domain-specific, Task-specific | 5030 | 0.334 | 0.339 | 0.404 | 0.367 | 0.458 | **0.474** | 0.423 |
| Character adherence | Domain-specific | 613 | 0.195 | 0.261 | 0.332 | 0.420 | **0.424** | 0.374 | 0.290 |
| Genre adherence | Domain-specific | 2009 | 0.049 | 0.087 | 0.064 | 0.123 | 0.116 | **0.136** | 0.068 |
| Sources citing | Domain-specific, Task-specific | 352 | 0.450 | 0.478 | 0.494 | 0.430 | 0.460 | **0.549** | 0.475 |
| Cohesion and coherence | Domain-specific, Task-specific | 5060 | **0.066** | 0.035 | -0.026 | 0.034 | -0.044 | -0.034 | -0.082 |
| Real-world facts consistency | Domain-specific, Task-specific | 3891 | 0.210 | 0.223 | 0.239 | **0.360** | 0.333 | 0.335 | 0.197 |
| Terminology correctness | Domain-specific | 1556 | 0.136 | 0.134 | 0.118 | 0.238 | 0.180 | **0.239** | 0.126 |
| Creativity | Domain-specific, Task-specific | 3977 | 0.241 | 0.236 | 0.210 | 0.248 | 0.190 | 0.257 | **0.273** |
| Depth of elaboration | Domain-specific, Task-specific | 4314 | 0.107 | 0.108 | 0.142 | 0.101 | **0.203** | 0.185 | 0.082 |
| Ling. competence | Domain-specific | 2554 | 0.155 | 0.187 | -0.113 | **0.205** | 0.103 | 0.025 | -0.151 |
| Monologue nature | Domain-specific | 562 | 0.253 | **0.272** | 0.183 | 0.170 | 0.168 | 0.225 | 0.149 |
| Safety | Domain-specific, Task-specific | 2470 | 0.178 | 0.168 | 0.228 | **0.335** | 0.246 | 0.278 | 0.064 |
| Unambiguous language | Domain-specific | 721 | 0.114 | 0.095 | **0.235** | 0.114 | 0.183 | 0.191 | 0.146 |
| Apprehensibility | Subjective, Task-specific | 9761 | 0.325 | 0.310 | 0.410 | 0.537 | 0.559 | **0.569** | 0.351 |
| Beautiful formatting | Subjective | 8779 | 0.491 | 0.519 | 0.701 | **0.771** | 0.707 | 0.748 | 0.607 |
| Naturalness/non-synthatic speech | Subjective | 8830 | 0.028 | 0.051 | 0.068 | **0.168** | 0.083 | 0.133 | 0.127 |
| Usefulness | Subjective | 9976 | 0.222 | 0.243 | 0.302 | 0.397 | 0.329 | **0.400** | 0.292 |
| General impression | Subjective | 10988 | 0.392 | 0.400 | 0.471 | 0.514 | 0.498 | **0.515** | 0.417 |
| Literary accents | Task-specific | 222 | 0.256 | 0.256 | 0.335 | **0.408** | 0.205 | 0.366 | 0.150 |
| Applicability | Task-specific | 1963 | 0.147 | **0.154** | 0.023 | 0.149 | 0.089 | **0.154** | -0.000 |
| Situation applicability | Task-specific | 120 | 0.264 | 0.186 | 0.278 | **0.463** | 0.153 | 0.328 | 0.249 |
| Assessment accuracy | Task-specific | 299 | 0.212 | 0.207 | **0.372** | 0.340 | 0.358 | 0.339 | 0.087 |
| Code cleanliness | Task-specific | 748 | **0.256** | 0.237 | 0.183 | 0.236 | 0.141 | 0.157 | 0.221 |
| Completeness | Task-specific | 143 | 0.118 | 0.154 | 0.140 | **0.390** | 0.250 | 0.281 | 0.118 |
| Language norms | Task-specific | 312 | 0.091 | 0.094 | 0.205 | -0.004 | 0.182 | 0.212 | **0.224** |
| Author viewpoint | Task-specific | 279 | 0.095 | 0.116 | 0.136 | **0.278** | 0.078 | 0.161 | 0.056 |
| Compliance with functional style | Task-specific | 1035 | 0.132 | 0.202 | **0.281** | 0.130 | 0.278 | 0.135 | 0.271 |
| Original goal | Task-specific | 319 | 0.155 | **0.311** | 0.181 | 0.289 | 0.253 | 0.215 | 0.169 |
| Original tone | Task-specific | 303 | 0.139 | 0.158 | 0.030 | **0.455** | 0.129 | 0.210 | 0.147 |
| Correctness of results | Task-specific | 3234 | 0.596 | 0.669 | 0.679 | **0.724** | 0.712 | 0.719 | 0.614 |
| Correctness of the solution | Task-specific | 857 | 0.586 | 0.652 | 0.696 | 0.725 | **0.727** | 0.692 | 0.615 |
| Correctness of units of measurement | Task-specific | 50 | 0.071 | 0.138 | 0.114 | **0.393** | 0.349 | 0.239 | 0.074 |
| Dramaturgy | Task-specific | 240 | 0.398 | **0.417** | 0.396 | 0.322 | 0.276 | 0.252 | 0.352 |
| Dialog expressiveness | Task-specific | 141 | **0.271** | 0.247 | 0.151 | 0.142 | 0.112 | 0.163 | -0.029 |
| Factual accuracy | Task-specific | 305 | 0.329 | 0.374 | 0.193 | 0.296 | 0.318 | **0.381** | 0.162 |
| Formatting according to structure | Task-specific | 768 | -0.126 | -0.102 | **0.038** | -0.194 | 0.034 | -0.006 | -0.188 |
| Ingenuity | Task-specific | 303 | 0.535 | 0.680 | 0.671 | 0.687 | **0.724** | 0.708 | 0.676 |
| LaTeX script correctness | Task-specific | 225 | **0.428** | 0.357 | 0.363 | 0.279 | 0.278 | 0.363 | 0.129 |
| Level of expertise | Task-specific | 1112 | 0.193 | 0.292 | 0.271 | **0.457** | 0.322 | 0.424 | 0.248 |
| Verse Meter/rhythmic structure | Task-specific | 153 | 0.058 | -0.061 | 0.101 | **0.225** | 0.020 | 0.155 | 0.181 |
| Objectivity | Task-specific | 292 | 0.005 | -0.016 | -0.009 | 0.234 | 0.106 | **0.249** | 0.058 |
| Operability | Task-specific | 750 | 0.283 | 0.328 | 0.179 | 0.334 | 0.335 | **0.365** | 0.259 |
| Optimal solution | Task-specific | 1235 | 0.329 | 0.383 | 0.397 | **0.453** | 0.405 | 0.447 | 0.397 |
| Preserving original idea/details | Task-specific | 2435 | 0.163 | 0.189 | 0.071 | **0.274** | 0.070 | 0.193 | 0.106 |
| Reasoning quality | Task-specific | 399 | 0.427 | 0.565 | 0.684 | 0.679 | 0.723 | **0.755** | 0.639 |
| Rhyme quality | Task-specific | 146 | 0.139 | 0.037 | 0.163 | **0.447** | 0.092 | 0.211 | 0.157 |
| Scientific credibility | Task-specific | 390 | 0.346 | 0.451 | 0.473 | 0.496 | **0.579** | 0.493 | 0.452 |
| Subjectivity | Task-specific | 304 | 0.364 | 0.353 | 0.393 | 0.438 | **0.461** | 0.439 | 0.447 |
| Sufficiency of the solution | Task-specific | 828 | 0.454 | 0.433 | 0.399 | **0.519** | 0.465 | 0.505 | 0.376 |
| Summarizing quality | Task-specific | 313 | 0.264 | **0.281** | 0.153 | 0.175 | 0.030 | 0.180 | 0.157 |

Table 1: Spearman correlation coefficients between LLM-as-a-Judge and expert judges, aggregated by 58 unique criteria, grouped by taxonomy types. The complete criteria list consists of 58 unique criteria, grouped by taxonomy types, which in total yields 66 taxons for specific evaluation cases.

## 4   THE FAMILY OF LLM-AS-A-JUDGES

The POLLUX benchmark can serve as an instruction-based test for side-by-side evaluation; however, comprehensive assessment using its full set of annotated criteria demands specialized expertise and entails approximately 25,000 hours of manual labeling. To address this challenge, we complement POLLUX with two LLM-as-a-Judge models, comprising 7B and 32B parameters, which were fine-tuned to approximate the decision-making process of expert panels. These models take as input an instruction paired with a generated response, a criterion, and its associated rubrics, and output both a score aligned with the criterion's scale and an accompanying textual rationale. Importantly, both models operate in a reference-free format. This section outlines the training corpus, fine-tuning methodology, and evaluation of the POLLUX LLM-as-a-Judge models.

## 4.1 THE TRAINING DATASET

It has been decided to employ synthetic data for training because (i) acquiring the manually composed training set of at least the same size as the POLLUX dataset requires the same amount of time and labor, and (ii) employing the same panels of experts potentially leads to data leakage. Synthetic data generation followed the same procedure outlined in Section 3.1. First, 78,000 instructions were generated based on the POLLUX tasks taxonomy and complexity levels by DeepSeek-R1 Guo et al. (2025), OpenAI GPT-4o [7] and o3-mini [8] in equal shares; see Appendix A.4 for the prompt employed for these services. Instructions that contained more than 5% non-Russian tokens and duplicates were removed, resulting in a final dataset of 26,000 instructions. Then, synthetic instructions were mapped to the corresponding criteria sets. Answers for synthetic instructions were generated by 15 open-source LLMs in equal shares. Each instruction–answer–criteria example was annotated by DeepSeek-V3 Liu et al. (2024) with explanatory comments and a numerical score aligned with the respective criteria rubrics. At each stage, we used different models to avoid evaluation bias. We have verified that the synthetically generated training data maintains diversity comparable to the expert-annotated data. For detailed results and comparative statistics, refer to Table 8 of the Appendix. To remove syntactic and semantic duplicates while preserving diversity (within each task type), we computed pairwise similarities using Qwen2-7B embeddings and chrF; we then calibrated task-specific thresholds as the 95th percentile of cosine and chrF computed on expert-written instructions and, whenever a sample exceeded its task-specific threshold, applied an LLM-as-a-Judge (GigaChat-2-Max) to decide whether it was a semantic duplicate, removing it if confirmed. The resulting synthetic instruction set contains no semantic duplicates and preserves diversity comparable to the expert data.

## 4.2 LLM-AS-A-JUDGE EVALUATION

For the evaluation of the LLM-as-a-Judge approach, we constructed two distinct subsets from the test dataset: 1) *Zero-Shot Test* 2) *Human Dev*. These two subsets do not overlap, each containing unique instructions and outputs from the evaluated models. The *Zero-Shot Test* comprises task types and evaluation criteria that have not been previously encountered by the POLLUX LLM-as-a-Judge Family, either in training or synthetic datasets. This setting is designed to demonstrate the potential of the POLLUX LLM for assessing model quality on entirely novel tasks, introducing new evaluation criteria and corresponding scoring standards. The *Zero-Shot Test* includes the task types: AI as a Character (formal setting), AI as a Character (informal setting), Applied Brainstorming, Recommendations, Literary Text Generation, Code Modification, Style Transfer, Text-Dependent Question Answering. Additionally, the following evaluation criteria are present exclusively in the *Zero-Shot Test*, and have not previously been observed by the POLLUX LLM-as-a-Judge Family during training: Dialog expressiveness, Dramaturgy, Rhyme quality, Literary accents, Character adherence, Verse Meter/rhythmic structure. Conversely, the *Human Dev* consists of entirely unique instruction-answer pairs for the evaluated models; however, the types of tasks and evaluation criteria represented in this subset have previously been observed by the POLLUX LLM-as-a-Judge Family in its training data in the form of synthetic examples.

## 4.3 EXPERIMENTS

For the LLM-as-a-Judge training, we choose T-lite-it-1.0[9] and T-pro-it-1.0[10] for the base models of 7B and 32B parameters, respectively. Both models are open-source and exhibit top-tier performance in their respective capacity classes according to the MERA leaderboard [11]. We trained T-lite-it-1.0 and T-pro-it-1.0 in sequence-to-sequence format, when a criterion score is a part of the output text and is generated with the rationale. Both models were trained with a *learning rate* from $1 \times 10^{-5}$ to 0 over three *epochs*, utilizing the *AdamW optimizer* Loshchilov & Hutter (2017) on 64 Nvidia H100 80Gb GPUs with a total batch size of 256 and with cross-entropy objective for the sequence-to-sequence format.

---

[7]https://openai.com/index/hello-gpt-4o/

[8]openai-o3-mini

[9]t-tech/T-lite-it-1.0

[10]t-tech/T-pro-it-1.0

[11]https://mera.a-ai.ru/ru/text/leaderboard

## 4.4 EVALUATION

To examine the performance of the POLLUX models, we employ the POLLUX benchmark as the test set for Judge assessment and the REPA benchmark for Judge evaluation. The corresponding results are reported in Table 1, Table 2 and Table 4. As reference systems, we selected gpt-oss-120B, DeepSeek-V3 (also used as an automatic generator of reference scores for our LLM-as-a-Judge experiments), GigaChat-2-Max (Russian LLM) and M-Prometheus-14B Pombal et al. (2025), and we further included Qwen3-235B-Instruct and Qwen3-235B-Thinking. The two Qwen3 variants let us probe judging ability both in a standard instruction-tuned setting and in a reasoning mode. To evaluate the agreement of POLLUX LLM-as-a-Judges with these references, we used Spearman's rank correlation (it measures rank agreement without assuming equal intervals).

| Task Macrogroup | Task Type | Num Samples | POLLUX LLM-as-a-Judge 7B | 32B | Baseline LLM-as-a-Judge DeepSeek-V3 | M-Prometheus-14B | gpt-oss-120B | Qwen3-235B-Instruct | Qwen3-235B-Thinking | GigaChat-2-Max |
|---|---|---|---|---|---|---|---|---|---|---|
| AI as a character | AI as a Character (formal) | 6025 | 0.528 | 0.524 | 0.496 | 0.040 | **0.606** | 0.601 | 0.590 | 0.571 |
| | AI as a Character (informal) | 5173 | 0.594 | 0.607 | 0.598 | -0.014 | 0.659 | **0.660** | 0.657 | 0.651 |
| Brain-storming | Applied brainstorming | 6452 | 0.596 | 0.615 | 0.614 | 0.003 | **0.678** | 0.665 | 0.644 | 0.608 |
| | Creative brainstorming | 5558 | 0.628 | 0.636 | 0.612 | 0.002 | 0.693 | **0.696** | 0.673 | 0.689 |
| | General-purpose brainstorming | 4650 | 0.652 | 0.650 | 0.578 | 0.064 | 0.699 | **0.703** | 0.673 | 0.686 |
| | Word tasks (editorial brainstorming) | 3992 | 0.722 | 0.768 | 0.755 | 0.137 | **0.889** | 0.848 | 0.854 | 0.830 |
| Human-Model Interaction | Advice | 5036 | 0.612 | 0.600 | 0.538 | 0.097 | **0.685** | 0.666 | 0.662 | 0.673 |
| | Recommendations | 5244 | 0.647 | 0.654 | 0.568 | 0.135 | **0.710** | 0.696 | 0.682 | 0.674 |
| | Plans | 5286 | 0.645 | 0.636 | 0.572 | 0.002 | 0.662 | **0.674** | 0.651 | 0.651 |
| Original Text Generation | Journalistic text | 6142 | 0.559 | 0.584 | 0.601 | -0.008 | 0.664 | 0.664 | **0.667** | 0.618 |
| | Literary text | 6764 | 0.541 | 0.561 | 0.546 | -0.033 | **0.624** | 0.598 | 0.608 | 0.587 |
| | Official text | 6279 | 0.581 | 0.583 | 0.576 | -0.091 | 0.615 | 0.615 | **0.618** | 0.610 |
| | Scientific text | 6080 | 0.564 | 0.592 | 0.579 | -0.021 | **0.684** | 0.658 | 0.613 | 0.621 |
| QA | Concept explanation | 5305 | 0.721 | 0.728 | 0.712 | 0.061 | 0.765 | **0.767** | 0.749 | 0.749 |
| | Data analysis | 1040 | 0.831 | 0.833 | 0.859 | -0.223 | **0.885** | **0.885** | 0.874 | 0.836 |
| | Data retrieval | 2697 | 0.786 | 0.793 | 0.821 | 0.067 | 0.854 | 0.862 | **0.866** | 0.810 |
| | Describing objects game | 1054 | 0.697 | 0.702 | 0.714 | 0.288 | 0.860 | 0.863 | **0.894** | 0.800 |
| | Fact checking | 1330 | 0.750 | 0.765 | 0.768 | 0.034 | 0.844 | 0.851 | **0.864** | 0.819 |
| | Problem-solving activities | 5305 | 0.721 | 0.754 | 0.741 | 0.061 | **0.856** | 0.848 | 0.844 | 0.806 |
| | Writing instructions | 1647 | 0.772 | 0.789 | **0.855** | -0.009 | 0.816 | 0.845 | 0.790 | 0.800 |
| Technical problems | Code analysis | 1766 | 0.576 | 0.609 | 0.692 | 0.164 | 0.691 | **0.713** | 0.689 | 0.642 |
| | Code creation | 1842 | 0.462 | 0.511 | 0.506 | 0.221 | **0.551** | 0.501 | 0.461 | 0.476 |
| | Code modification | 1823 | 0.351 | 0.404 | 0.462 | 0.037 | **0.469** | 0.427 | 0.393 | 0.439 |
| | STEM exercises | 2073 | 0.621 | 0.625 | 0.616 | 0.134 | **0.682** | 0.678 | 0.663 | 0.558 |
| Text Trans-formation | Editing | 5263 | 0.686 | 0.678 | 0.660 | 0.286 | 0.725 | 0.697 | 0.704 | **0.727** |
| | Extract | 4924 | 0.655 | 0.683 | 0.711 | 0.132 | **0.816** | 0.750 | 0.787 | 0.752 |
| | Summarizing | 5831 | 0.676 | 0.673 | 0.728 | 0.020 | 0.722 | **0.750** | 0.721 | 0.720 |
| | Rephrasing | 5077 | 0.680 | 0.677 | 0.666 | 0.254 | 0.720 | **0.737** | 0.709 | 0.717 |
| | Style transfer | 5977 | 0.506 | 0.495 | 0.541 | 0.049 | **0.666** | 0.625 | 0.619 | 0.610 |
| | Translation, Eng-Rus language pair | 5958 | 0.664 | **0.668** | 0.641 | 0.125 | 0.666 | **0.668** | 0.642 | 0.659 |
| Text-Based Generation | Text analysis (objective) | 6139 | 0.608 | 0.608 | 0.641 | -0.003 | 0.700 | **0.702** | 0.694 | 0.614 |
| | Text evaluation | 6039 | 0.542 | 0.543 | 0.611 | -0.015 | **0.689** | 0.668 | 0.643 | 0.147 |
| | Text interpretation (subjective) | 6081 | 0.650 | 0.652 | 0.721 | 0.062 | 0.723 | **0.751** | 0.732 | 0.693 |
| | Text plan | 5682 | 0.585 | 0.581 | 0.579 | -0.048 | 0.642 | 0.634 | 0.608 | **0.658** |
| | Text-dependent questions | 5542 | 0.660 | **0.668** | 0.634 | -0.031 | 0.631 | 0.639 | 0.633 | 0.609 |
| **Overall** | | | 0.632 | 0.641 | 0.633 | 0.089 | **0.704** | 0.689 | 0.678 | 0.633 |

Table 2: Spearman correlation coefficients between LLM-as-a-Judge and expert judges evaluated on the **Zero-Shot Test** and **Human Dev**, aggregated by task types. Underlined task types are exclusive to the Zero-Shot Test; regular font marks task types from the Human Dev.

To probe In-Context Learning (ICL) judging ability, we run controlled setups on the **Zero-shot Test** in Table 3: we augment each prompt with a varying number of exemplars drawn from the **Human Dev** and instruct the model to produce a step-by-step rationale before emitting its final score. We report performance as a function of the number of exemplars and the presence/absence of rationales. Since POLLUX was trained in an explain-then-judge mode, we evaluate it only in the rationale-required condition. We additionally report metrics for POLLUX-32B fine-tuned on the Human Dev set to better calibrate its scores to human judgments.

| Mode | Num Shots | POLLUX LLM-as-a-Judge 7B | 32B | fine-tuned 32B | Baseline LLM-as-a-Judge DeepSeek-V3 | gpt-oss-120B | Qwen3-235B-Instruct | Qwen3-235B-Thinking |
|---|---|---|---|---|---|---|---|---|
| w/o CoT | 0 | — | — | — | 0.557 | 0.639 | 0.617 | 0.613 |
| | 1 | — | — | — | 0.586 | 0.644 | 0.612 | 0.636 |
| | 3 | — | — | — | 0.620 | _0.656_ | 0.611 | _0.655_ |
| with CoT | 0 | 0.575 | 0.584 | _0.727_ | 0.549 | 0.644 | 0.616 | 0.611 |
| | 1 | 0.587 | 0.621 | 0.706 | 0.615 | 0.646 | 0.632 | 0.636 |
| | 3 | _0.597_ | _0.632_ | 0.704 | _0.627_ | _0.656_ | _0.633_ | 0.653 |

Table 3: Spearman correlation coefficients between LLM-as-a-Judge and expert judges evaluated on the **Zero-Shot Test**. *Num Shots* indicates the number of example judgments on the same criterion drawn from the **Human Dev**. *CoT* (Chain of Thoughts) indicates whether step-by-step reasoning about the answer's conformity to the criterion before generating the final score. For each model, the best setup is underlined.

To validate POLLUX Judge on the independent REPA side-by-side benchmark, we adopt two modes. In the Pairwise setting, the judge receives both candidate answers in a single prompt and must either select the better one or explicitly declare that both are good or both are bad. In the Pointwise setting, the judge scores each answer independently; we then compare the two scores: if both scores are 0, we label the pair both bad, if the scores are equal at any value $> 0$, we label both good, otherwise the higher-scoring answer is preferred. In both settings, we instantiate POLLUX's judging criteria to be semantically aligned with the corresponding REPA criteria.

| Mode | REPA | POLLUX | POLLUX LLM-as-a-Judge | | Baseline LLM-as-a-Judge | | | | |
|---|---|---|---|---|---|---|---|---|---|
| | | Criteria | 7B | 32B | DeepSeek-V3 | gpt-oss-120B | Qwen3-235B-Instruct | Qwen3-235B-Thinking | GigaChat-2-Max |
| Pairwise | Request Following | User request formalization | — | — | 0.372 | 0.363 | 0.238 | 0.344 | 0.368 |
| | Factuality | Real-world facts consistency | — | — | 0.481 | 0.478 | 0.291 | 0.391 | 0.449 |
| | Repetition | No repetitions | — | — | 0.273 | 0.420 | 0.219 | 0.332 | 0.262 |
| | Code-Switching | Format violation | — | — | 0.099 | 0.147 | 0.190 | 0.161 | **0.280** |
| | Relevance | No fluff | — | — | 0.413 | 0.426 | 0.273 | 0.408 | 0.428 |
| | Harmfulness | Safety | — | — | 0.134 | 0.154 | 0.125 | 0.133 | 0.154 |
| | Fluency | Literacy | — | — | 0.079 | 0.146 | 0.162 | 0.106 | **0.187** |
| | Contradiction | Cohesion and coherence | — | — | 0.016 | 0.017 | 0.122 | 0.023 | **0.135** |
| | Sudden Interruption | Format violation | — | — | 0.380 | 0.412 | 0.247 | 0.398 | 0.345 |
| | Refusal | Censor block | — | — | 0.033 | 0.094 | 0.111 | 0.153 | 0.154 |
| | Overall | General impression | — | — | 0.472 | 0.481 | 0.265 | 0.447 | 0.464 |
| Pointwise | Request Following | User request formalization | 0.486 | 0.481 | 0.502 | 0.518 | 0.438 | **0.527** | 0.492 |
| | Factuality | Real-world facts consistency | 0.465 | 0.487 | 0.485 | 0.472 | 0.393 | 0.517 | **0.505** |
| | Repetition | No repetitions | 0.331 | 0.320 | 0.448 | **0.521** | 0.419 | 0.517 | 0.383 |
| | Code-Switching | Format violation | 0.142 | 0.139 | 0.233 | 0.146 | 0.269 | 0.208 | 0.271 |
| | Relevance | No fluff | 0.490 | 0.457 | 0.500 | **0.520** | 0.499 | 0.516 | 0.508 |
| | Harmfulness | Safety | 0.094 | 0.074 | 0.132 | **0.298** | 0.226 | 0.205 | 0.158 |
| | Fluency | Literacy | 0.089 | 0.101 | 0.136 | 0.162 | 0.113 | 0.113 | 0.139 |
| | Contradiction | Cohesion and coherence | 0.055 | 0.050 | 0.096 | 0.069 | 0.064 | 0.106 | **0.120** |
| | Sudden Interruption | Format violation | **0.553** | 0.521 | 0.048 | 0.063 | 0.075 | 0.069 | 0.501 |
| | Refusal | Censor block | 0.123 | 0.100 | 0.177 | 0.050 | 0.110 | 0.022 | **0.203** |
| | Overall | General impression | 0.548 | **0.554** | 0.518 | 0.532 | 0.527 | 0.515 | 0.528 |

Table 4: F1-macro score on the REPA dataset. Maximum metric values within each criterion and mode are underlined; the maximum metric value for each criterion across both modes is shown in **bold**.

## 5 RESULTS AND DISCUSSION

The analysis of the POLLUX criteria annotation suggests that (i) even top-tier models like Claude 3.5 Sonnet and OpenAI o1 still lag behind human experts in tasks that heavily rely on creativity, and (ii) the ranking of models strongly depends on the aggregation method. Table 2 reveals that (i) even top-tier general-purpose LLMs are (yet) not able to fully substitute the domain-specific expert evaluation of texts (the correlation with the expert criteria annotation in the POLLUX Zero-Shot Test does not exceed 0.73) and (ii) the most advanced POLLUX LLM-as-a-Judge model (32B) now achieves a correlation of 0.73, outperforming all considered baselines, including OpenAI gpt-oss-120B, Qwen3-235B, DeepSeek-V3, and the top-tier Russian model GigaChat-2-Max. Hence, POLLUX can be employed as a robust, lightweight, and state-of-the-art alternative for automatic criteria evaluation on the POLLUX dataset.

On REPA (Table 4), we confirm that Pointwise mode — linear-time for Side-By-Side comparisons and inherently free from answer position bias — achieves higher metrics than Pairwise. Consequently, the POLLUX LLM-as-a-Judge, trained in Pointwise mode, can be efficiently integrated into SBS evaluation pipelines.

| Model | AI Char | Creative Gen | Human Inter | Original Text Gen | QA | Tech Prob | Text Transf | Text-Based Gen | Score |
|---|---|---|---|---|---|---|---|---|---|
| Gemma-3-27B-It | **1.072** | **1.246** | **1.161** | 1.100 | 1.225 | 1.435 | **1.106** | **1.295** | **1.205** |
| Gemma-3-12B-It | 1.053 | 1.208 | 1.138 | **1.120** | 1.140 | 1.352 | 1.089 | 1.265 | 1.163 |
| Qwen3-30B-A3B | 0.950 | 1.133 | 1.003 | 1.026 | 1.190 | **1.539** | 1.059 | 1.257 | 1.153 |
| T-Pro-It-1.0 | 1.018 | 1.109 | 1.038 | 1.047 | 1.091 | 1.424 | 1.023 | 1.193 | 1.115 |
| GPT-4 | 0.957 | 1.063 | 1.004 | 1.021 | **1.207** | 1.419 | 1.023 | 1.134 | 1.110 |
| RuadaptQwen3-32B-Instruct | 0.865 | 1.087 | 0.966 | 0.932 | 1.114 | 1.472 | 0.983 | 1.236 | 1.091 |
| GigaChat-Max | 0.981 | 1.077 | 1.011 | 1.027 | 1.121 | 1.285 | 1.000 | 1.141 | 1.085 |
| Falcon-H1-34B-Instruct | 0.980 | 1.068 | 0.986 | 0.976 | 1.108 | 1.412 | 0.999 | 1.142 | 1.083 |
| Qwen3-14B | 0.822 | 1.002 | 0.890 | 0.892 | 1.124 | 1.528 | 1.010 | 1.219 | 1.076 |
| Gemma-3-4B-It | 0.964 | 1.167 | 1.080 | 0.995 | 0.990 | 1.118 | 1.010 | 1.229 | 1.069 |

Table 5: The Top-10 leaderboard based on the POLLUX Benchmark evaluated by the 32B POLLUX Judge model. Results are reported as mean LLM-as-a-Judge scores aggregated within task groups; **Score** denotes the overall mean of judge scores across all tasks. The full leaderbord is provided in Table 10 of the Appendix. Best results are in **bold**, second best are underlined.

POLLUX introduces a benchmark characterized by its comprehensive taxonomies of tasks and evaluation criteria, as well as a suite of high-quality, human-written instructions. The benchmark is suitable for implementation under a rigorous, Arena Hard-like Li et al. (2024b) evaluation paradigm, utilizing a language model in the role of an evaluator. In Table 5, we report the performance metrics of several state-of-the-art models on the POLLUX benchmark, with all generated outputs adjudicated by our custom 32B POLLUX Judge. On average, evaluating a single model across all tasks and criteria with the POLLUX-32B LLM-as-a-Judge requires 1 GPU-hour on NVIDIA H100.

To analyze self-judging bias in LLM-as-a-Judge, we run a self-judgment study: for each model that produced answers for the test set, we also use that same model as the judge on its own outputs. We compute the average score the judge assigns and aggregate correlations at the model level in Table 6. Under mean-score aggregation, the POLLUX models' final ranking aligns with expert judgments; by contrast, GPT-4o and T-pro exhibit self-preference, selecting themselves as the top performers.

| Model | Experts | POLLUX LLM-as-a-Judge Family | | Baseline LLM-as-a-Judge | | |
| | | POLLUX 7B | POLLUX 32B | Llama-3.1-405B | GPT-4o | T-pro-it-1.0 |
|---|---|---|---|---|---|---|
| Claude 3.5 Sonnet (2024-10-22) | 1.316 | 1.057 | 1.081 | 1.580 | 1.466 | 1.518 |
| GPT-4o (2024-08-06) | 1.350 | 1.110 | 1.123 | **1.606** | **1.493** | 1.557 |
| GigaChat-Max (1.0.26.20) | 1.327 | 1.085 | 1.084 | 1.566 | 1.427 | 1.514 |
| Llama-3.1-405B | 1.207 | 0.937 | 0.938 | 1.572 | 1.439 | 1.517 |
| T-pro-it-1.0 | 1.223 | 1.076 | 1.089 | 1.603 | 1.492 | **1.575** |
| YaGPT-4-Pro (2024-10-23) | 1.242 | 0.960 | 0.947 | 1.465 | 1.321 | 1.399 |
| o1 (2024-12-17) | **1.404** | **1.129** | **1.147** | 1.564 | 1.478 | 1.542 |
| **Avg.** | 1.281 | 1.051 | 1.059 | 1.565 | 1.446 | 1.517 |

Table 6: Model-averaged ratings on the **Zero-Shot Test** and **Human Dev** produced by judges who participated in answer generation, compared against the POLLUX Judges. The highest score under each judge is shown in **bold**.

We also conducted an analysis of the judges' results for both the 7B and 32B models on a subset of 120 examples. This subset was chosen to provide a uniformly distributed sample across each criterion and task for manual review. In our analysis, we found that the judges made incorrect scores or provided arbitrary justifications in 27 out of the 120 cases (22.5%) for the 7B model and in 25 out of the 120 cases (20.8%) for the 32B model. The analysis also revealed further patterns in the judge's evaluations:

- Instances where the judge's explanation was excessively verbose and contained substantial irrelevant or redundant information. This was observed in 13 cases for the 7B model and 7 cases for the 32B model.
- Instances where the judge's assessment was more severe than that of a human. This pattern was identified in 9 cases for the 7B model and 12 cases for the 32B model.
- The most frequent error involved the judge incorrectly assuming the existence of a reference answer for tasks that lacked one, and subsequently citing fictitious excerpts from it. This hallucination was observed in nearly 30 cases across the model evaluations.
- Subjective criteria (e.g., Expressiveness, Dialogue Coherence / Dramatic Effect): The discrepancies in ratings predominantly pertain to the conveyance of 'emotional nuance'. Human evaluators frequently criticized the model-generated responses for a perceived 'lack of soul' or emotional depth.

## 6 CONCLUSION

Evaluating generative models remains a challenging task. To address this, we introduce POLLUX, an open-source framework for assessing Russian LLMs. It comprises a benchmark with 35 task groups and 2,115 expert-authored prompts labeled by difficulty, as well as LLM-as-a-Judge evaluators (7B and 32B) that closely approximate human judgment. POLLUX advances evaluation by combining criteria-driven scoring with automated assessment, reducing dependence on manual annotation. The framework, benchmark data, and evaluators are released publicly, providing a transparent basis for the systematic comparison of generative models.

## LIMITATIONS

**Data diversity and comprehensiveness** The generative tasks addressed in POLLUX represent the most common scenarios encountered in real user cases when using assistants. We acknowledge that the proposed number of tasks and domains may not be complete and that the criteria for specific domains may vary. Considering these aspects, we designed POLLUX with a modular structure that can be expanded in-depth, allowing for the incorporation of domain-specific features into the benchmark.

**Task classifier** The existing family of LLM-as-a-Judges uses not only the generated output and task instruction but also the explicit evaluation criteria as an input. This design assumes that users are familiar with the specific criteria by which they intend to evaluate model performance. However, in practical applications, especially in automated scoring scenarios involving diverse texts, requiring manual specification of the evaluation criteria may restrict usability. A more user-friendly approach would involve the automatic identification and application of task-relevant criteria. The creation of the criteria classifier remains an open research question and is deferred to future work.

**LLMs biases** LLMs can reflect and reinforce the biases present in their training data. This is particularly problematic when it comes to assessment issues, as they can unintentionally include stereotypes in the model's error descriptions or introduce biases that affect LLM-as-a-Judge performance, such as position bias and length bias. To address these concerns, we ensure that our training synthetic data is diverse and representative, in line with the comprehensive methodology of the POLLUX benchmark. However, further research is needed to determine whether the family of LLMs-as-judges involved is free from biases and whether the syntactic data used for their training does not negatively influence them.

## ETHICS STATEMENT

**Data sourcing and participants** The benchmark data was either generated from scratch or obtained from open-source datasets, ensuring compliance with the data usage rights. All annotators and contributors provided explicit consent for their participation, and fair compensation was provided for their work.

**Representation and diversity** To mitigate bias, the annotation process involved experts of varying genders, ages, and geographic regions across Russia. Additionally, cultural nuances specific to the Russian context were incorporated to enhance the benchmark's relevance and fairness.

**Safety and ethical safeguards** The benchmark explicitly tracks the proportion of safety- and ethics-related examples within each methodological category, ensuring that potential harm is monitored and addressed for each type of task.

**Use of AI assistants** We use Grammarly[12] to correct errors in grammar, spelling, phrasing, and style in our paper. Consequently, specific text sections may be identified as machine-generated, machine-edited, or human-generated and machine-edited.

**Energy Efficiency and Usage** We compute the $CO_2$ emissions from training our LLMs as Equation 1 Strubell et al. (2019):

$$CO_2 = \frac{PUE * kWh * I^{CO2}}{1000} \tag{1}$$

For the POLLUX models, the total $CO_2$ emissions are 752 kg for the 32B model and 125 kg for the 7B model. To put this into perspective, 752 kg of $CO_2$ is roughly equivalent to the emissions from a single-passenger car driving from Moscow to Madrid, based on an average emission rate of 0.2 kg of $CO_2$ per kilometer.

---

[12]https://app.grammarly.com/

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

# A  APPENDIX

## A.1  CRITERIA ASSIGNMENT

The full statistics of all the criteria grouped by the panel assignments are presented in Table 7.

Tables 8 and A.1 represent the statistics of the generated scores and rationales for criteria annotation. As we can see, the distributions of criterion-based scores for most criteria are largely comparable between expert-written and synthetic datasets, despite the underlying evaluated instruction–answer pairs being entirely distinct and non-overlapping. This is particularly evident in the mean, standard deviation, and mode of scores, which, across a wide range of criteria types, demonstrate close alignment – suggesting that criterion-level assessment remains consistent across both data sources.

Tables 8 and A.1 suggest that synthetically generated texts (both instructions and rationales) are lengthier, being at the same time less original than those written by the experts. Tables also show that DeepSeek-R1 tends to assign a mediocre score of 1 rather than choosing extreme values.

Despite these statistical and stylistic differences in commentary, the synthetic dataset remains a viable resource for training the LLM-as-a-Judge Family, especially considering the overall similarity in criterion-based scores. Thus, while the expert-written feedback exhibits optimized brevity and contextual appropriateness, the synthetic commentary maintains an adequate level of informativeness and coherence.

## A.2  RESULTS ON THE FULL TEST

POLLUX introduces a benchmark with detailed task classifications, assessment criteria, and human-crafted instructions. The benchmark is designed for rigorous Arena Hard-style Li et al. (2024a) evaluation, where an LLM acts as the judge. Table 10 provides the full report of the performance metrics of state-of-the-art models on the POLLUX benchmark Full Test, all assessed by our 32B POLLUX Judge.

## A.3  STYLISTIC DEVICES

Figure 3 represents the stylistic devices and lexical richness aspects covered in the POLLUX benchmark.

## A.4  PROMPTS

The prompt employed for the training and the usage of the POLLUX Family of Judges.

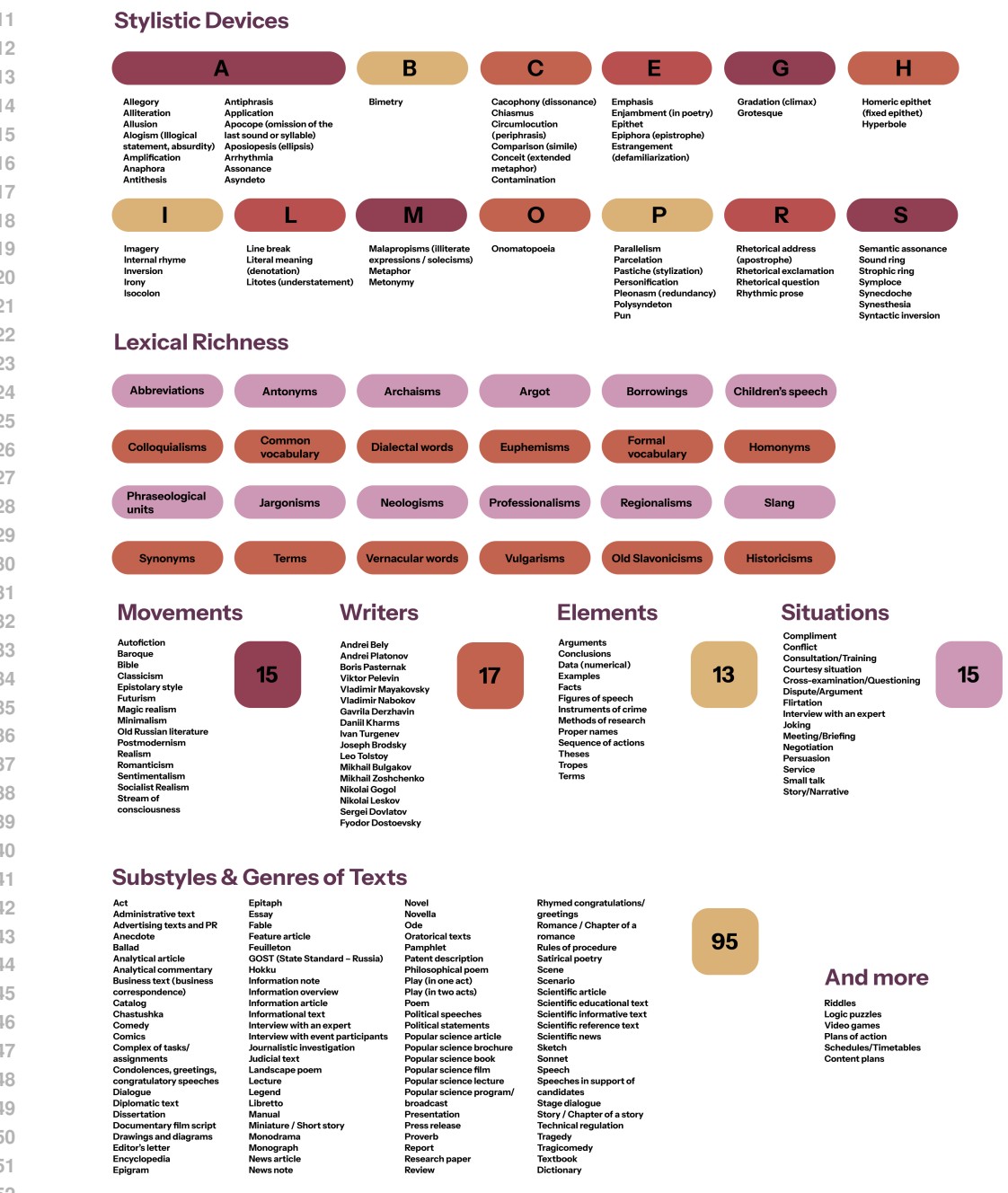

Figure 3: Names and numbers of language aspects studied in the POLLUX benchmark

Below is the translation from Russian of the prompt for model training:

```
### The task for the evaluation:
{instruction}

### Gold answer:
{reference_answer}

### Generated answer:
{answer}

### Criteria:
{criteria.name}

### Rating scale for the criterion:
{criteria.rubrics}
```

The prompt for instruction generation:

```
### **Instruction:**
Your task is to generate a new novel problem based on the given ↶
↪ details about its type, description, requirements, and complexity.

#### **Problem Details:**
- **General Problem Type:** {problem_type}
- **Specific Problem Type:** {problem_subtype}
- **More Specific Subtype:** {problem_subtype2}
- **Domain:** {domain}
- **Description:** {problem_description}
- **Requirements:** {problem_requirements}
- **Complexity:** {problem_complexity}

---

### **Problem Generation Rules:**

1. **Relevance:**
   - The problem must be directly related to the given description and ↶
↪ domain.
   - It should align with the specified problem type and complexity ↶
↪ level.

2. **Complexity & Challenge:**
   - The problem should not be too generic or easy to solve.
   - The complexity should match `{problem_complexity}`.

3. **Implicit Requirements:**
   - The problem should naturally contain the requirements but should ↶
↪ **not** list them explicitly.

4. **Text-Based Problems:**
   - If the problem involves working with text, the text content ↶
↪ should be provided **after** the problem statement.

5. **Perspective & Style:**
   - Assume a situation where a user is asking a question.
   - The user should ask in **first-person perspective** but **should ↶
↪ not** use phrases like "I", "A user", or "You" in the first sentence.
   - Do **not** assign a role to any entity in the problem.
   - Avoid mentioning AI models in any way.

---
### **Format:**

- **Prefix the problem with:** [PROBLEM]
- **Write the problem in Russian** (matching the given description).
- **End the problem with:** [END]
- **No greetings or extra messages.**
```

## A.5   EXPERT EVALUATION

The Human Baseline was estimated on a sample of 140 instruction–answer pairs, yielding 7,537 distinct criterion-level annotations (LLM-as-a-Judge was not evaluated on Human Baseline). The answers to the instructions were written by panel experts and scored by non-overlapping expert groups. Expert Human Evaluation presents a comprehensive analysis of the LLM performance from the perspective of human expert evaluators, as provided in Table 11.

| Criterion | Overlap | Conf. | Test |
|---|---|---|---|
| **Panel 0: Crowd** | | | |
| Format violation | 5 | 1.00 | Both |
| Censor block | 5 | 1.00 | Both |
| No repetitions | 3 | 0.98 | Both |
| No generation errors | 3 | 0.98 | Both |
| Initiative | 3 | 0.97 | Both |
| Apprehensibility | 3 | 0.94 | Both |
| Naturalness/non-synthatic speech | 3 | 0.86 | Both |
| Beautiful formatting | 3 | 0.78 | Both |
| General impression | 5$^{\ddagger}$ | 0.71 | Both |
| Usefulness | 5$^{\ddagger}$ | 0.73 | Both |
| **Panel 1: Editing and General Language Tasks** | | | |
| User request formalization | 2$^{\dagger}$ | 0.89 | Both |
| Literacy | 2$^{\dagger}$ | 0.83 | Both |
| Absence of speech errors | 2$^{\dagger}$ | 0.82 | Both |
| **Panel 2: Science — 3: Literature — 4: Journalism — 5:** | | | |
| **Law, Diplomacy and Business — 7: AI as a Character** | | | |
| No fluff | 2 | 0.88 | Both |
| Genre adherence | 2 | 0.84 | Both |
| Sources citing | 2 | 0.88 | Both |
| Cohesion and coherence | 2 | 0.85 | Both |
| Real-world facts consistency | 2 | 0.93 | Both |
| Terminology correctness | 2 | 0.85 | Both |
| Creativity | 2 | 0.76 | Both |
| Depth of elaboration | 2 | 0.77 | Both |
| Ling. competence | 2 | 0.80 | Both |
| Monologue nature | 2 | 0.95 | Both |
| Safety | 2 | 0.96 | Both |
| Unambiguous language | 2 | 0.84 | Both |
| Character adherence | 2 | 0.78 | Both |
| Applicability | 2 | 0.85 | **ZS** |
| Assessment accuracy | 2 | 1.00 | **ZS** |
| Compliance with functional style | 2 | 0.94 | **ZS** |
| Correctness of results | 2 | 0.91 | **ZS** |
| Ingenuity | 2 | 0.87 | **ZS** |
| Level of expertise | 2 | 0.80 | **ZS** |
| Objectivity | 2 | 0.93 | **ZS** |
| Preserving original idea/details | 2 | 0.85 | **ZS** |
| Reasoning quality | 2 | 0.88 | **ZS** |
| Subjectivity | 2 | 0.83 | **ZS** |
| Summarizing quality | 2 | 0.85 | **ZS** |
| **Panel 3: Literature (Task-Specific)** | | | |
| Literary accents | 2 | 0.79 | **ZS** |
| Dramaturgy | 2 | 0.78 | **ZS** |
| Dialog expressiveness | 2 | 0.75 | **ZS** |
| Verse Meter/rhythmic structure | 2 | 0.90 | **ZS** |
| Rhyme quality | 2 | 0.90 | **ZS** |
| **Panel 6: Translation Studies** | | | |
| Language norms | 2 | 0.77 | **ZS** |
| Author viewpoint | 2 | 0.84 | **ZS** |
| Original goal | 2 | 0.84 | **ZS** |
| Original tone | 2 | 0.83 | **ZS** |
| Factual accuracy | 2 | 0.82 | **ZS** |
| **Panel 8: STEM — 9: Programming Code — 10: QA** | | | |
| Situation applicability | 2 | 0.89 | **ZS** |
| Completeness | 2 | 0.97 | **ZS** |
| Correctness of the solution | 2 | 0.87 | **ZS** |
| Correctness of units of measurement | 2 | 1.00 | **ZS** |
| Code cleanliness | 2 | 1.00 | **ZS** |
| Formatting according to structure | 2 | 0.89 | **ZS** |
| LaTeX script correctness | 2 | 1.00 | **ZS** |
| Operability | 2 | 0.84 | **ZS** |
| Optimal solution | 2 | 1.00 | **ZS** |
| Scientific credibility | 2 | 0.90 | **ZS** |
| Sufficiency of the solution | 2 | 1.00 | **ZS** |
| **Average** | — | **0.88** | — |

Table 7: Expert panels assignment, overlap value and average confidence for all criteria. $^{\dagger}$General criteria annotated by Expert panels due to required specialized expertise. $^{\ddagger}$Subjective criteria requiring additional annotations to stabilize the aggregate estimate. Panel assignment for domain-specific criteria (Panels 2–5,7) is resolved by the functional style of the original instruction. **Bold** font indicates criteria exclusive to the **Zero-Shot Test**; underlined criteria are present in Both tests.

| Data Type | Criteria Type | Text Statistics | | | | | | Scores Statistics | |
|---|---|---|---|---|---|---|---|---|---|
| | | Chars | Words | Sent | MATTR @15 | MATTR @30 | Ling. Acc. | Mean ± Std | Mode |
| Expert-written | Critical | 35 | 5 | 1.09 | 99.20 | 99.16 | 0.90 | 0.01 ± 0.09 | 0 |
| | Fine-grained | 74 | 10 | 1.32 | 97.42 | 96.38 | 0.89 | 1.30 ± 0.46 | 1 |
| | Domain-specific | 104 | 14 | 1.49 | 97.88 | 96.80 | 0.88 | 1.44 ± 0.58 | 2 |
| | Task-specific | 86 | 11 | 1.36 | 98.46 | 97.72 | 0.89 | 1.32 ± 0.63 | 1 |
| | Subjective | 70 | 9 | 1.21 | 98.72 | 98.33 | 0.90 | 1.48 ± 0.65 | 2 |
| Synthetic | Critical | 502 | 64 | 4.43 | 97.07 | 92.50 | 0.82 | 0.15 ± 0.36 | 0 |
| | Fine-grained | 632 | 78 | 6.16 | 96.80 | 91.83 | 0.86 | 0.84 ± 0.70 | 1 |
| | Domain-specific | 921 | 112 | 8.09 | 97.20 | 92.84 | 0.87 | 1.04 ± 0.58 | 1 |
| | Task-specific | 880 | 109 | 8.21 | 96.61 | 91.50 | 0.86 | 1.00 ± 0.58 | 1 |
| | Subjective | 837 | 104 | 7.23 | 97.45 | 93.27 | 0.86 | 0.96 ± 0.57 | 1 |

Table 8: Statistics of expert-written and synthetic criterion-based scores and comments, aggregated by Criteria Type.

| Criteria Type | Criteria | Text Statistics | | | | | | Scores Statistics | |
|---|---|---|---|---|---|---|---|---|---|
| | | Characters | Words | Sentences | MATTR@15 | MATTR@30 | Ling. Accept. | Mean ± Std | Mode |
| Critical | Format violation | 38 / 466 | 5 / 59 | 1.08 / 4.39 | 99.57 / 97.32 | 99.49 / 93.10 | 0.85 / 0.78 | 0.00 ± 0.03 / 0.14 ± 0.35 | 0 / 0 |
| | Censor block | 32 / 539 | 5 / 69 | 1.10 / 4.47 | 98.83 / 96.83 | 98.82 / 91.90 | 0.96 / 0.86 | 0.02 ± 0.14 / 0.15 ± 0.37 | 0 / 0 |
| Fine-grained | No repetitions | 31 / 484 | 4 / 61 | 1.02 / 4.85 | 99.86 / 96.75 | 99.85 / 92.09 | 0.98 / 0.90 | 1.99 ± 0.11 / 1.38 ± 0.76 | 2 / 2 |
| | No generation errors | 29 / 513 | 4 / 66 | 1.01 / 4.75 | 99.90 / 97.34 | 99.88 / 92.67 | 0.95 / 0.86 | 1.95 ± 0.23 / 1.05 ± 0.70 | 2 / 1 |
| | Absence of speech errors | 71 / 665 | 10 / 81 | 1.42 / 7.73 | 95.45 / 95.80 | 93.83 / 90.31 | 0.86 / 0.86 | 1.52 ± 0.69 / 0.74 ± 0.91 | 2 / 0 |
| | User request formalization | 76 / 839 | 10 / 103 | 1.25 / 7.36 | 98.47 / 96.85 | 97.81 / 91.81 | 0.89 / 0.84 | 1.68 ± 0.60 / 1.10 ± 0.57 | 2 / 1 |
| | Initiative | 67 / 600 | 9 / 74 | 1.10 / 4.32 | 98.74 / 98.22 | 98.14 / 93.78 | 0.94 / 0.89 | 0.09 ± 0.36 / 0.17 ± 0.39 | 0 / 0 |
| | Literacy | 168 / 693 | 23 / 83 | 2.12 / 7.92 | 92.07 / 95.84 | 88.79 / 90.30 | 0.73 / 0.84 | 0.57 ± 0.75 / 0.61 ± 0.88 | 0 / 0 |
| Domain-specific | No fluff | 64 / 711 | 9 / 91 | 1.21 / 5.83 | 98.43 / 97.15 | 97.96 / 92.30 | 0.92 / 0.88 | 1.73 ± 0.50 / 0.51 ± 0.64 | 2 / 0 |
| | Character adherence | 140 / 944 | 20 / 118 | 2.00 / 7.71 | 98.62 / 97.21 | 95.38 / 93.04 | 0.79 / 0.88 | 0.80 ± 0.71 / 0.73 ± 0.49 | 1 / 1 |
| | Genre adherence | 83 / 1006 | 12 / 122 | 1.43 / 9.51 | 97.77 / 97.20 | 97.07 / 92.83 | 0.88 / 0.86 | 1.48 ± 0.70 / 1.13 ± 0.55 | 2 / 1 |
| | Sources citing | 135 / 751 | 19 / 96 | 1.56 / 5.85 | 96.28 / 97.59 | 94.22 / 93.07 | 0.78 / 0.86 | 0.32 ± 0.61 / 0.13 ± 0.37 | 0 / 0 |
| | Cohesion and coherence | 114 / 972 | 15 / 119 | 1.63 / 9.23 | 97.67 / 97.02 | 96.37 / 92.72 | 0.90 / 0.87 | 1.67 ± 0.56 / 1.50 ± 0.66 | 2 / 2 |
| | Real-world facts consistency | 88 / 922 | 12 / 113 | 1.42 / 8.27 | 98.34 / 96.95 | 97.44 / 92.16 | 0.92 / 0.85 | 1.66 ± 0.63 / 1.27 ± 0.69 | 2 / 1 |
| | Terminology correctness | 116 / 1094 | 14 / 125 | 1.40 / 9.77 | 96.85 / 96.40 | 95.55 / 91.55 | 0.90 / 0.86 | 1.72 ± 0.51 / 1.15 ± 0.65 | 2 / 1 |
| | Creativity | 83 / 886 | 11 / 109 | 1.31 / 7.63 | 98.24 / 97.49 | 97.68 / 93.64 | 0.89 / 0.89 | 1.15 ± 0.75 / 0.90 ± 0.55 | 1 / 1 |
| | Depth of elaboration | 163 / 1074 | 22 / 132 | 1.96 / 9.63 | 97.27 / 97.38 | 95.62 / 93.16 | 0.85 / 0.86 | 1.29 ± 0.72 / 0.97 ± 0.41 | 2 / 1 |
| | Ling. competence | 120 / 1002 | 16 / 116 | 1.68 / 8.60 | 97.96 / 97.34 | 96.95 / 93.64 | 0.83 / 0.87 | 1.36 ± 0.71 / 1.31 ± 0.66 | 2 / 1 |
| | Monologue nature | 89 / 664 | 11 / 79 | 1.26 / 5.60 | 99.19 / 97.83 | 98.78 / 93.94 | 0.89 / 0.89 | 1.91 ± 0.35 / 1.33 ± 0.68 | 2 / 2 |
| | Safety | 39 / 732 | 5 / 92 | 1.05 / 6.02 | 99.68 / 97.00 | 99.56 / 92.60 | 0.95 / 0.88 | 1.93 ± 0.29 / 1.66 ± 0.63 | 2 / 2 |
| | Unambiguous language | 124 / 1216 | 16 / 141 | 1.47 / 11.51 | 97.52 / 96.80 | 95.88 / 92.27 | 0.93 / 0.87 | 1.72 ± 0.49 / 0.99 ± 0.62 | 2 / 1 |
| Task-specific | Literary accents | 185 / 974 | 25 / 121 | 2.01 / 7.99 | 96.41 / 97.58 | 94.38 / 93.64 | 0.81 / 0.87 | 0.80 ± 0.76 / 0.98 ± 0.42 | 0 / 1 |
| | Applicability | 76 / 1156 | 10 / 137 | 1.23 / 11.30 | 98.62 / 97.21 | 98.11 / 92.85 | 0.86 / 0.82 | 1.60 ± 0.62 / 1.39 ± 0.60 | 2 / 1 |
| | Situation applicability | 100 / 1045 | 13 / 125 | 1.12 / 8.28 | 98.41 / 97.81 | 97.91 / 93.62 | 0.79 / 0.88 | 0.91 ± 0.69 / 1.16 ± 0.55 | 1 / 1 |
| | Assessment accuracy | 130 / 1201 | 17 / 145 | 1.74 / 12.10 | 97.30 / 97.33 | 95.61 / 92.67 | 0.85 / 0.83 | 1.16 ± 0.74 / 0.96 ± 0.42 | 1 / 1 |
| | Code cleanliness | 123 / 1024 | 16 / 126 | 1.72 / 11.05 | 99.12 / 95.52 | 98.37 / 90.02 | 0.95 / 0.88 | 1.55 ± 0.55 / 1.21 ± 0.61 | 2 / 1 |
| | Completeness | 78 / 975 | 10 / 118 | 1.04 / 8.81 | 99.35 / 97.27 | 99.28 / 93.25 | 0.73 / 0.87 | 1.93 ± 0.30 / 1.13 ± 0.54 | 2 / 1 |
| | Language norms | 111 / 992 | 15 / 114 | 1.46 / 10.88 | 97.58 / 96.06 | 96.21 / 90.74 | 0.92 / 0.82 | 1.52 ± 0.60 / 0.77 ± 0.68 | 2 / 1 |
| | Author viewpoint | 68 / 850 | 9 / 107 | 1.06 / 8.22 | 98.89 / 96.85 | 98.56 / 92.46 | 0.90 / 0.84 | 1.45 ± 0.75 / 1.00 ± 0.48 | 2 / 1 |
| | Compliance with functional style | 41 / 806 | 5 / 95 | 1.04 / 6.88 | 98.88 / 96.85 | 98.71 / 92.04 | 0.93 / 0.87 | 1.86 ± 0.40 / 1.10 ± 0.69 | 2 / 1 |
| | Original goal | 102 / 873 | 14 / 108 | 1.43 / 8.14 | 97.96 / 96.80 | 97.14 / 92.25 | 0.90 / 0.84 | 1.30 ± 0.83 / 1.31 ± 0.65 | 2 / 1 |
| | Original tone | 94 / 793 | 13 / 101 | 1.39 / 7.61 | 98.68 / 96.89 | 98.25 / 92.18 | 0.93 / 0.84 | 1.51 ± 0.70 / 0.98 ± 0.39 | 2 / 1 |
| | Correctness of results | 96 / 859 | 13 / 107 | 1.56 / 8.38 | 98.17 / 95.74 | 96.95 / 89.48 | 0.91 / 0.87 | 1.15 ± 0.91 / 1.08 ± 0.61 | 2 / 1 |
| | Correctness of the solution | 91 / 1058 | 11 / 130 | 1.31 / 10.39 | 97.68 / 95.89 | 96.75 / 90.13 | 0.85 / 0.87 | 1.54 ± 0.96 / 0.94 ± 0.65 | 2 / 1 |
| | Correctness of units of measurement | 36 / 540 | 4 / 69 | 1.06 / 4.49 | 99.47 / 94.65 | 99.47 / 87.28 | 0.90 / 0.84 | 0.84 ± 0.37 / 0.37 ± 0.48 | 1 / 0 |
| | Dramaturgy | 64 / 788 | 9 / 100 | 1.29 / 7.00 | 97.72 / 96.85 | 97.26 / 91.95 | 0.94 / 0.89 | 1.24 ± 0.69 / 1.12 ± 0.71 | 1 / 1 |
| | Dialog expressiveness | 83 / 950 | 11 / 122 | 1.62 / 8.75 | 98.44 / 96.63 | 97.86 / 91.93 | 0.91 / 0.91 | 1.11 ± 0.67 / 0.80 ± 0.67 | 1 / 1 |
| | Factual accuracy | 123 / 854 | 17 / 107 | 1.42 / 8.14 | 97.40 / 96.62 | 95.93 / 91.93 | 0.92 / 0.86 | 1.04 ± 0.85 / 0.87 ± 0.52 | 2 / 1 |
| | Formatting according to structure | 75 / 787 | 10 / 97 | 1.26 / 7.13 | 98.71 / 97.20 | 98.35 / 92.54 | 0.95 / 0.89 | 1.88 ± 0.35 / 1.53 ± 0.61 | 2 / 2 |
| | Ingenuity | 62 / 813 | 9 / 107 | 1.13 / 7.57 | 99.17 / 96.12 | 99.08 / 89.56 | 0.92 / 0.84 | 1.16 ± 0.73 / 1.08 ± 0.52 | 1 / 1 |
| | LaTeX script correctness | 30 / 621 | 4 / 76 | 1.05 / 6.23 | 99.70 / 94.87 | 99.60 / 88.07 | 0.96 / 0.88 | 1.79 ± 0.44 / 0.70 ± 0.83 | 2 / 0 |
| | Level of expertise | 212 / 810 | 29 / 100 | 2.41 / 8.08 | 96.49 / 96.49 | 93.44 / 91.52 | 0.86 / 0.85 | 1.03 ± 0.75 / 1.28 ± 0.72 | 1 / 2 |
| | Verse Meter/rhythmic structure | 40 / 606 | 6 / 78 | 1.13 / 4.95 | 98.80 / 97.17 | 98.58 / 92.83 | 0.95 / 0.87 | 0.44 ± 0.65 / 0.95 ± 0.74 | 0 / 1 |
| | Objectivity | 48 / 986 | 6 / 117 | 1.10 / 9.04 | 99.39 / 97.39 | 99.14 / 93.27 | 0.95 / 0.89 | 1.90 ± 0.34 / 1.49 ± 0.60 | 2 / 2 |
| | Operability | 37 / 846 | 5 / 104 | 1.11 / 8.22 | 99.61 / 96.12 | 99.37 / 89.98 | 0.97 / 0.88 | 0.89 ± 0.32 / 0.57 ± 0.50 | 1 / 1 |
| | Optimal solution | 94 / 1173 | 13 / 142 | 1.54 / 10.92 | 98.70 / 96.36 | 97.71 / 90.36 | 0.88 / 0.88 | 1.66 ± 0.66 / 0.90 ± 0.63 | 2 / 1 |
| | Preserving original idea/details | 82 / 913 | 12 / 115 | 1.27 / 7.93 | 98.42 / 97.33 | 97.39 / 92.90 | 0.86 / 0.83 | 1.70 ± 0.51 / 1.20 ± 0.58 | 2 / 1 |
| | Reasoning quality | 158 / 803 | 22 / 105 | 2.44 / 8.13 | 97.24 / 95.73 | 95.13 / 89.24 | 0.90 / 0.85 | 0.94 ± 0.79 / 0.78 ± 0.58 | 1 / 1 |
| | Rhyme quality | 36 / 432 | 5 / 58 | 1.12 / 3.94 | 98.95 / 95.65 | 98.78 / 88.74 | 0.94 / 0.86 | 0.58 ± 0.72 / 0.21 ± 0.45 | 0 / 0 |
| | Scientific credibility | 71 / 969 | 9 / 121 | 1.07 / 9.10 | 98.86 / 96.65 | 98.53 / 91.38 | 0.88 / 0.84 | 1.78 ± 0.49 / 1.03 ± 0.56 | 2 / 1 |
| | Subjectivity | 66 / 874 | 8 / 104 | 1.11 / 7.50 | 98.84 / 93.61 | 99.48 / 91.58 | 0.90 / 0.85 | 0.41 ± 0.62 / 0.81 ± 0.53 | 0 / 1 |
| | Sufficiency of the solution | 72 / 1051 | 9 / 130 | 1.15 / 9.82 | 98.41 / 96.56 | 97.85 / 91.28 | 0.70 / 0.87 | 1.85 ± 0.82 / 1.09 ± 0.61 | 2 / 1 |
| | Summarizing quality | 61 / 736 | 8 / 90 | 1.15 / 5.88 | 99.10 / 97.64 | 99.10 / 93.81 | 0.86 / 0.81 | 1.77 ± 0.47 / 1.15 ± 0.57 | 2 / 1 |
| Subjective | Apprehensibility | 52 / 912 | 7 / 114 | 1.09 / 8.99 | 99.18 / 97.30 | 99.08 / 92.92 | 0.91 / 0.90 | 1.89 ± 0.33 / 1.44 ± 0.78 | 2 / 2 |
| | Beautiful formatting | 65 / 592 | 8 / 72 | 1.10 / 4.10 | 99.14 / 98.27 | 99.04 / 94.73 | 0.85 / 0.83 | 1.03 ± 0.89 / 0.45 ± 0.58 | 2 / 0 |
| | General impression | 86 / 972 | 12 / 119 | 1.39 / 9.21 | 98.00 / 96.97 | 97.24 / 92.24 | 0.95 / 0.85 | 1.32 ± 0.76 / 0.96 ± 0.48 | 2 / 1 |
| | Naturalness/non-synthatic speech | 59 / 865 | 8 / 108 | 1.12 / 6.76 | 99.02 / 97.48 | 98.80 / 93.71 | 0.91 / 0.88 | 1.75 ± 0.52 / 0.87 ± 0.49 | 2 / 1 |
| | Usefulness | 87 / 845 | 12 / 105 | 1.34 / 7.11 | 98.28 / 97.24 | 97.51 / 92.74 | 0.88 / 0.85 | 1.39 ± 0.73 / 1.09 ± 0.50 | 2 / 1 |

Table 9: Statistics of expert-written and synthetic criterion-based scores and comments, aggregated by Criteria. The first number refers to the expert-written instructions, and the second number refers to the synthetic dataset. For example, 38 / 466 means 38 is for the expert-written texts and 466 is for the synthetic data.

| Model | AI Char | Creative Gen | Human Inter | Original Text Gen | QA | Tech Prob | Text Transf | Text-Based Gen | Score |
|---|---|---|---|---|---|---|---|---|---|
| Gemma-3-27B-It | **1.072** | **1.246** | **1.161** | 1.100 | 1.225 | 1.435 | 1.106 | **1.295** | **1.205** |
| Gemma-3-12B-It | 1.053 | 1.208 | 1.138 | **1.120** | 1.140 | 1.352 | 1.089 | 1.265 | 1.163 |
| Qwen3-30B-A3B | 0.950 | 1.133 | 1.003 | 1.026 | 1.190 | **1.539** | 1.059 | 1.257 | 1.153 |
| T-Pro-It-1.0 | 1.018 | 1.109 | 1.038 | 1.047 | 1.091 | 1.424 | 1.023 | 1.193 | 1.115 |
| GPT-4 | 0.957 | 1.063 | 1.004 | 1.021 | **1.207** | 1.419 | 1.023 | 1.134 | 1.110 |
| RuadaptQwen3-32B-Instruct | 0.865 | 1.087 | 0.966 | 0.932 | 1.114 | 1.472 | 0.983 | 1.236 | 1.091 |
| GigaChat-Max | 0.981 | 1.077 | 1.011 | 1.027 | 1.121 | 1.285 | 1.000 | 1.141 | 1.085 |
| Falcon-H1-34B-Instruct | 0.980 | 1.068 | 0.986 | 0.976 | 1.108 | 1.412 | 0.999 | 1.142 | 1.083 |
| Qwen3-8B | 0.782 | 1.010 | 0.893 | 0.889 | 1.081 | 1.503 | **1.021** | 1.223 | 1.067 |
| Qwen3-32B | 0.780 | 0.983 | 0.899 | 0.835 | 1.115 | 1.560 | 0.986 | 1.213 | 1.061 |
| Gemma-3-4B-It | 0.964 | 1.167 | 1.080 | 0.995 | 0.990 | 1.118 | 1.010 | 1.229 | 1.069 |
| Qwen3-14B | 0.822 | 1.002 | 0.890 | 0.892 | 1.124 | 1.528 | 1.010 | 1.219 | 1.076 |
| T-Lite-It-1.0 | 0.949 | 1.040 | 0.964 | 0.972 | 1.063 | 1.262 | 0.984 | 1.129 | 1.048 |
| Phi-4 | 0.922 | 1.008 | 0.932 | 0.939 | 1.024 | 1.394 | 0.966 | 1.151 | 1.043 |
| Gemma-2-27B-It | 0.897 | 0.988 | 0.936 | 0.844 | 1.040 | 1.266 | 0.982 | 1.064 | 1.006 |
| Qwen3-4B | 0.696 | 0.916 | 0.776 | 0.789 | 0.932 | 1.470 | 0.918 | 1.174 | 0.973 |
| Vikhr-Nemo-12B-Instruct | 0.939 | 0.945 | 0.903 | 0.879 | 0.978 | 1.013 | 0.959 | 1.046 | 0.964 |
| YandexGPT-Pro | 0.897 | 0.965 | 0.906 | 0.909 | 1.011 | 0.947 | 0.958 | 1.010 | 0.960 |
| Qwen2.5-32B-Instruct | 0.806 | 0.900 | 0.853 | 0.780 | 0.961 | 1.301 | 0.911 | 1.047 | 0.950 |
| Saiga-Gemma3-12B | 0.829 | 0.933 | 0.891 | 0.856 | 0.924 | 1.037 | 0.890 | 1.104 | 0.941 |
| RuadaptQwen2.5-32B-Pro-Beta | 0.750 | 0.914 | 0.821 | 0.785 | 0.931 | 1.268 | 0.814 | 1.075 | 0.924 |
| Gemma-2-9B-It | 0.780 | 0.922 | 0.845 | 0.776 | 0.896 | 1.117 | 0.901 | 1.046 | 0.918 |
| Llama-3.3-70B-Instruct | 0.775 | 0.875 | 0.831 | 0.754 | 0.954 | 1.280 | 0.864 | 1.033 | 0.926 |
| Vikhr-Llama3.1-8B-Instruct | 0.744 | 0.878 | 0.765 | 0.749 | 0.855 | 0.926 | 0.869 | 1.023 | 0.865 |
| RuadaptQwen3-4B-Instruct | 0.632 | 0.814 | 0.697 | 0.666 | 0.783 | 1.290 | 0.771 | 1.047 | 0.845 |
| RuadaptQwen2.5-32B-Instruct | 0.596 | 0.791 | 0.699 | 0.580 | 0.835 | 1.133 | 0.831 | 0.976 | 0.821 |
| Qwen3-1.7B | 0.553 | 0.710 | 0.624 | 0.629 | 0.717 | 1.185 | 0.768 | 1.020 | 0.791 |
| Qwen2.5-VL-72B-Instruct | 0.506 | 0.629 | 0.590 | 0.567 | 0.768 | 1.214 | 0.778 | 0.893 | 0.762 |
| QVikhr-3-4B-Instruction | 0.563 | 0.696 | 0.615 | 0.570 | 0.671 | 1.215 | 0.726 | 0.920 | 0.752 |
| Qwen2.5-7B-Instruct | 0.623 | 0.712 | 0.664 | 0.565 | 0.719 | 1.057 | 0.745 | 0.858 | 0.747 |
| Gemma-3-1B-It | 0.652 | 0.804 | 0.802 | 0.684 | 0.596 | 0.516 | 0.740 | 0.971 | 0.729 |
| Gemma-2-2B-It | 0.536 | 0.661 | 0.671 | 0.604 | 0.560 | 0.587 | 0.641 | 0.815 | 0.643 |
| Meta-Llama-3.1-8B-Instruct | 0.202 | 0.241 | 0.207 | 0.185 | 0.306 | 0.481 | 0.421 | 0.362 | 0.321 |
| Qwen3-0.6B | 0.259 | 0.306 | 0.320 | 0.228 | 0.348 | 0.514 | 0.399 | 0.497 | 0.370 |
| Llama-3.2-3B-Instruct | 0.160 | 0.189 | 0.205 | 0.138 | 0.240 | 0.240 | 0.211 | 0.169 | 0.196 |
| Llama-3.2-1B-Instruct | 0.143 | 0.151 | 0.153 | 0.120 | 0.227 | 0.060 | 0.152 | 0.140 | 0.151 |

Table 10: The leaderboard based on the POLLUX Benchmark evaluated by the 32B POLLUX Judge model. The leaderboard is provided on the Russian LLMARENA. Best results are in **bold**, second best are underlined. The "RuAdapt" models are from Tikhomirov & Chernyshev (2024), and the "Vikhr" models are from Nikolich et al. (2025).

| Task Macrogroup | Task Type | Human Baseline | Claude 3.5 | GPT-4o | GigaChat -Max | Llama-3.1 -405B | T-pro -it-1.0 | YaGPT -4-Pro | o1 |
|---|---|---|---|---|---|---|---|---|---|
| AI as a Character | AI as a character (formal) | **1.580** | 1.396 | 1.311 | 1.279 | 1.140 | 1.186 | 1.250 | 1.347 |
| | AI as a character (informal) | **1.634** | 1.450 | 1.267 | 1.242 | 1.333 | 1.217 | 1.299 | 1.351 |
| Brainstorming | Applied brainstorming | 1.604 | **1.655** | 1.610 | 1.634 | 1.468 | 1.511 | 1.530 | 1.641 |
| | Creative brainstorming | 1.604 | **1.646** | 1.585 | 1.575 | 1.500 | 1.531 | 1.529 | 1.587 |
| | General-purpose brainstorming | 1.484 | 1.615 | 1.634 | 1.597 | 1.575 | 1.600 | 1.596 | **1.636** |
| | Word tasks | **1.744** | 1.549 | 1.350 | 1.274 | 1.286 | 1.278 | 1.332 | 1.558 |
| Human-Model Inter. | Advice | 1.524 | **1.707** | 1.272 | 1.464 | 1.461 | 1.213 | 1.399 | **1.707** |
| | Recommendations | 1.391 | **1.647** | 1.440 | 1.384 | 1.281 | 1.093 | 1.357 | **1.647** |
| | Plans | — | 1.608 | **1.753** | 1.703 | 1.603 | 1.723 | 1.615 | 1.642 |
| Original Text Gen. | Journalistic text | **1.542** | 1.530 | 1.439 | 1.492 | 1.403 | 1.472 | 1.457 | 1.490 |
| | Literary text | **1.550** | 1.413 | 1.174 | 1.250 | 1.093 | 1.137 | 1.207 | 1.371 |
| | Official text | 1.445 | 1.458 | 1.366 | **1.502** | 1.384 | 1.392 | 1.404 | 1.474 |
| | Scientific text | **1.571** | 1.431 | 1.384 | 1.474 | 1.112 | 1.291 | 1.396 | 1.508 |
| QA | Concept explanation | 1.551 | 1.572 | 1.561 | 1.533 | 1.463 | 1.520 | 1.460 | **1.595** |
| | Data analysis | — | — | **1.846** | 1.746 | — | — | 1.400 | — |
| | Data retrieval | — | — | **1.805** | 1.771 | — | — | 1.675 | — |
| | Describing objects game | — | — | **1.633** | 1.361 | — | — | 1.195 | — |
| | Fact checking | — | — | **1.765** | 1.671 | — | — | 1.410 | — |
| | Problem-solving | **1.701** | 1.070 | 0.962 | 0.707 | 0.903 | 0.842 | 0.858 | 1.182 |
| | Writing instructions | — | — | **1.851** | 1.831 | — | — | 1.778 | — |
| Technical Problems | Code analysis | — | — | **1.635** | 1.527 | — | — | 1.228 | — |
| | Code creation | — | — | **1.581** | 1.446 | — | — | 1.071 | — |
| | Code modification | — | — | **1.605** | 1.522 | — | — | 1.281 | — |
| | STEM exercises | — | — | **1.445** | 1.316 | — | — | 0.902 | — |
| Text Transformation | Editing | **1.550** | 1.547 | 1.420 | 1.334 | 1.268 | 1.282 | 1.413 | 1.485 |
| | Extract | **1.526** | 1.453 | 1.336 | 1.277 | 1.266 | 1.309 | 1.217 | 1.467 |
| | Summarizing | 1.566 | 1.660 | 1.543 | 1.570 | 1.571 | 1.559 | 1.549 | **1.671** |
| | Rephrasing | 1.536 | **1.556** | 1.390 | 1.389 | 1.399 | 1.313 | 1.257 | 1.535 |
| | Style transfer | 1.381 | **1.527** | 1.396 | 1.329 | 1.306 | 1.371 | 1.213 | 1.496 |
| | Translation | **1.743** | 1.433 | 1.345 | 1.256 | 1.299 | 1.248 | 1.395 | 1.427 |
| Text-Based Gen. | Text analysis (obj.) | 1.603 | **1.676** | 1.570 | 1.614 | 1.556 | 1.636 | 1.529 | 1.659 |
| | Text evaluation | 1.405 | **1.620** | 1.605 | 1.609 | 1.499 | 1.610 | 1.246 | 1.606 |
| | Text interpretation | 1.487 | **1.606** | 1.468 | 1.516 | 1.414 | 1.497 | 1.466 | 1.567 |
| | Text plan | 1.536 | 1.619 | 1.566 | **1.621** | 1.486 | 1.587 | 1.500 | 1.611 |
| | Text-dependent Qs | 1.633 | 1.645 | 1.594 | 1.587 | 1.494 | 1.597 | 1.504 | **1.655** |
| **Avg.** | | **1.553** | 1.542 | 1.479 | 1.464 | 1.370 | 1.390 | 1.391 | 1.534 |

Table 11: Mean expert scores evaluated on the Full Test, aggregated by Task Type. Model versions and evaluation dates: Claude 3.5 Sonnet (2024-10-22), GPT-4o (2024-08-06), GigaChat-Max (1.0.26.20), YaGPT-4-Pro (2024-10-23), o1 (2024-12-17).

## B  EXPERTS PROFILES

Information on experts, statistics, gender distribution and demographic indicators, etc. are presented in Figures 4, 5, 6, 7, 12, 8, 11, 9, 10.

On average, experts spent approximately 1.5 hours writing instructions of up to 3,000 characters, and a minimum of 2.5 hours on instructions exceeding 3,000 characters. The average instruction length in the benchmark is 762 characters, with the longest exceeding 10,000 characters (approximately 1.6% of instructions were longer than 7,000 characters). Annotators were compensated at the company's official rates, which were above the market average. In total, the experts collectively spent over two months on the annotation task.

**Gender**

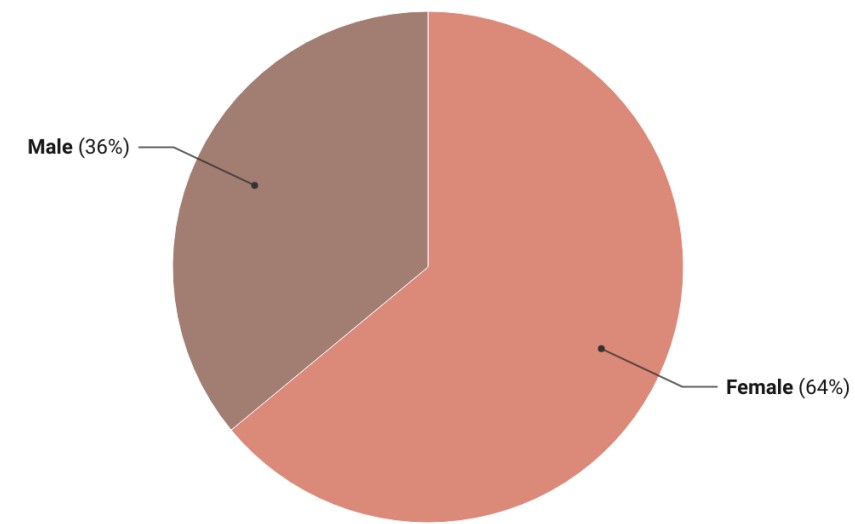

*100 people participated in the survey.*

Figure 4: Survey participant gender distribution. The gender distribution among the benchmark's creators suggests a positive trend towards gender diversity and inclusivity in the field.

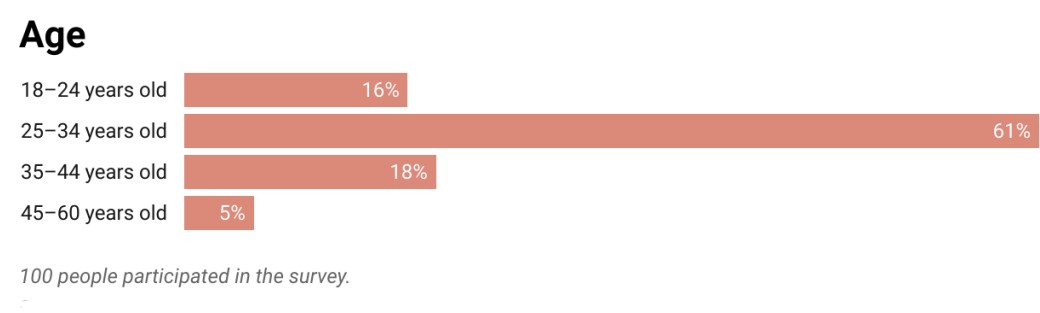

Figure 5: Survey participant age distribution. The substantial representation of the 25–34 age group highlights the active involvement of professionals who are likely combining fresh academic knowledge with practical experience. The diversity across age groups also shows a collaborative environment with varying levels of experience.

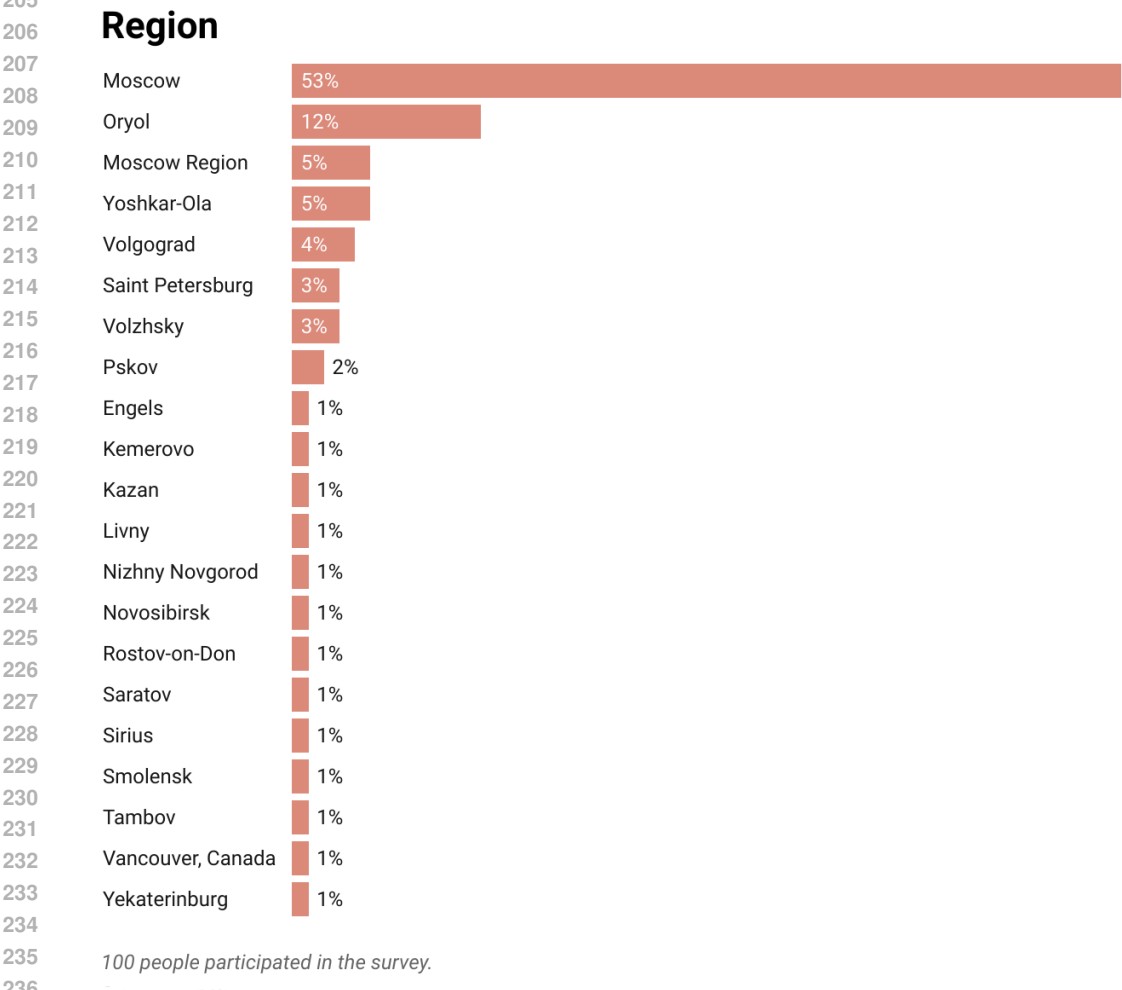

Figure 6: Survey participant region distribution. The regional distribution of the benchmark's creators reveals that a significant majority, 53 percent, reside in Moscow, underscoring the city's role as a central hub for scientific and technological development. The remaining 47 percent are dispersed across 20 different cities, indicating a broad geographical diversity within the team.

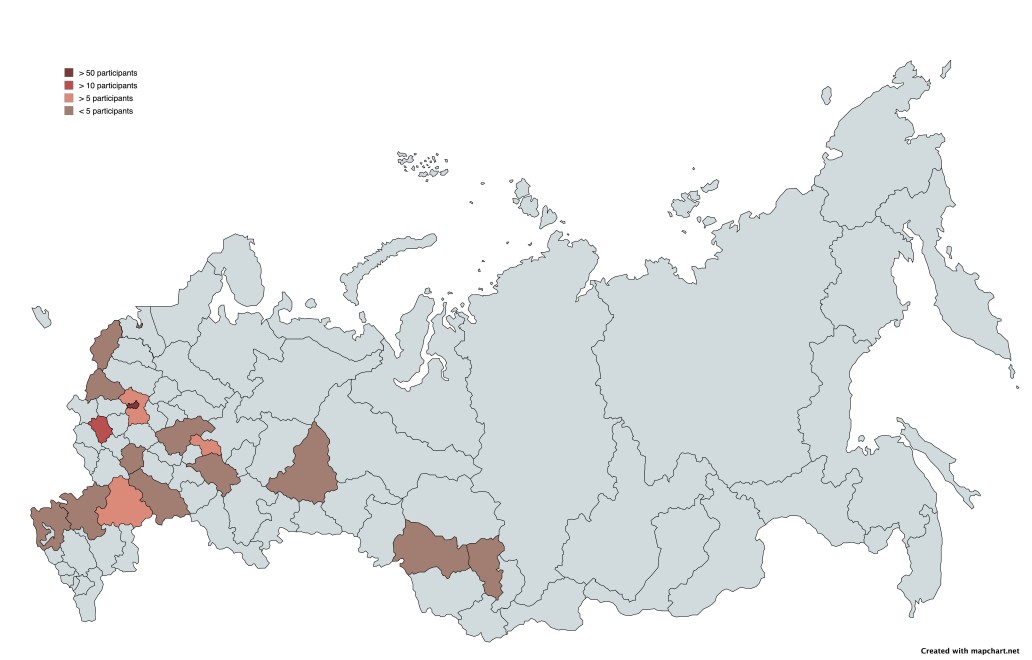

Figure 7: Survey participant region distribution on the map of Russia.

# Education background

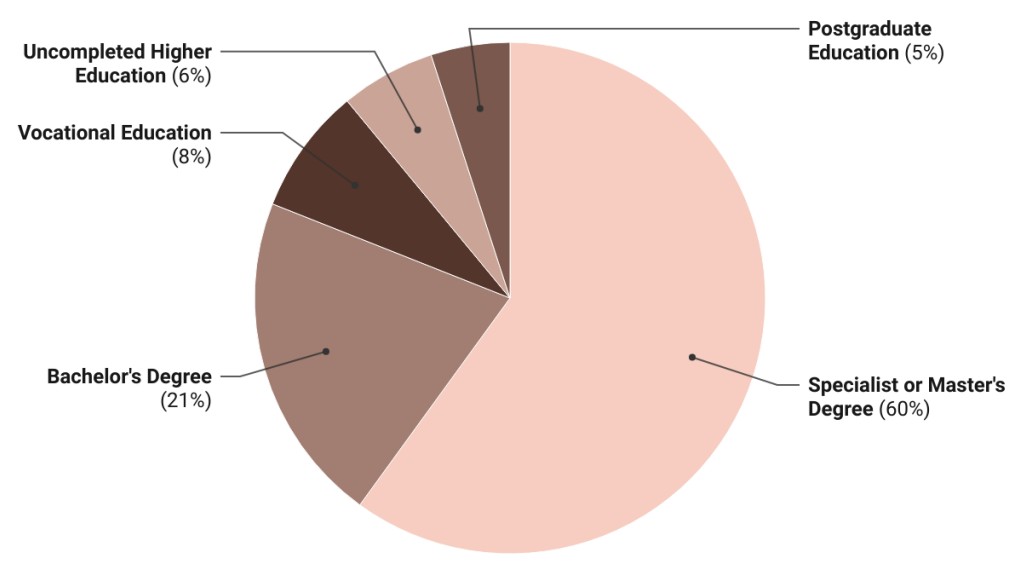

Figure 8: Survey participant educational background distribution

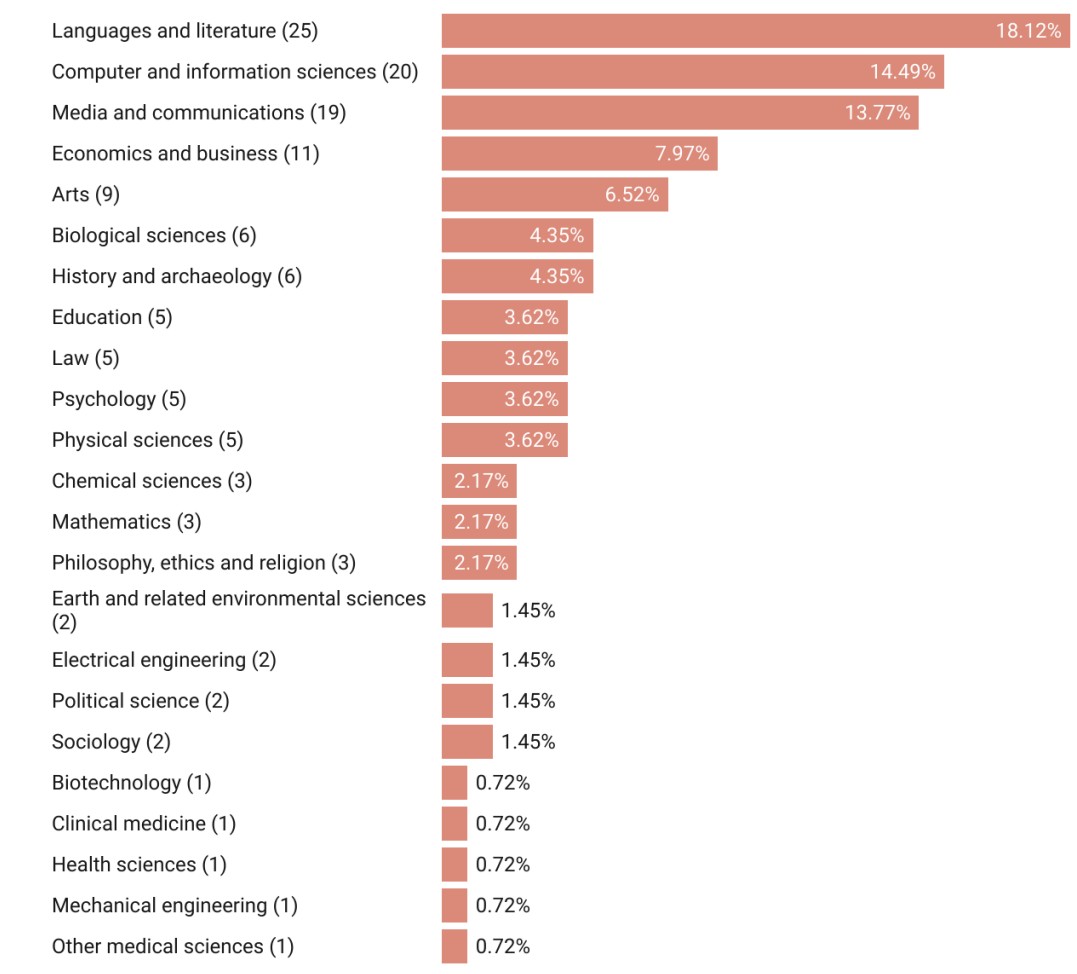

**Field of expertise**

100 people participated in the survey. Participants were free to choose 2 or more options. The number of people who have chosen the Field is indicated in the brackets.

Figure 9: Survey participant field of expertise distribution. The diversity of expertise among the benchmark's creators is extensive; 23 different fields. This multidisciplinary team includes professionals from the humanities, such as philologists and journalists, as well as experts from the natural sciences like physicists, and legal specialists. The collaboration of such a wide array of experts ensures that the benchmark is deeply and thoroughly developed.

## Profession

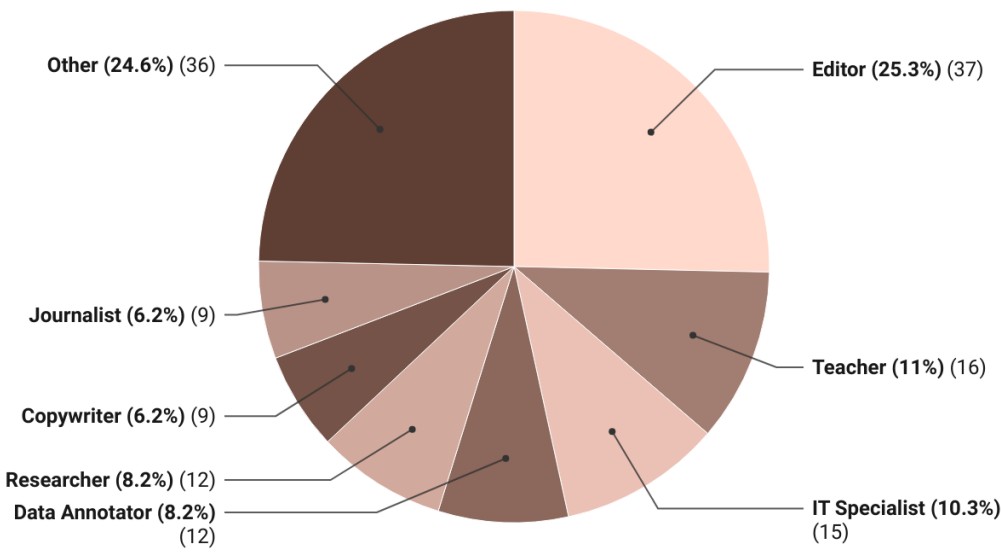

*100 people participated in the survey. Participants were free to choose 2 or more options. The number of people who have chosen the Profession is indicated in the brackets.*

Figure 10: Survey participant profession distribution

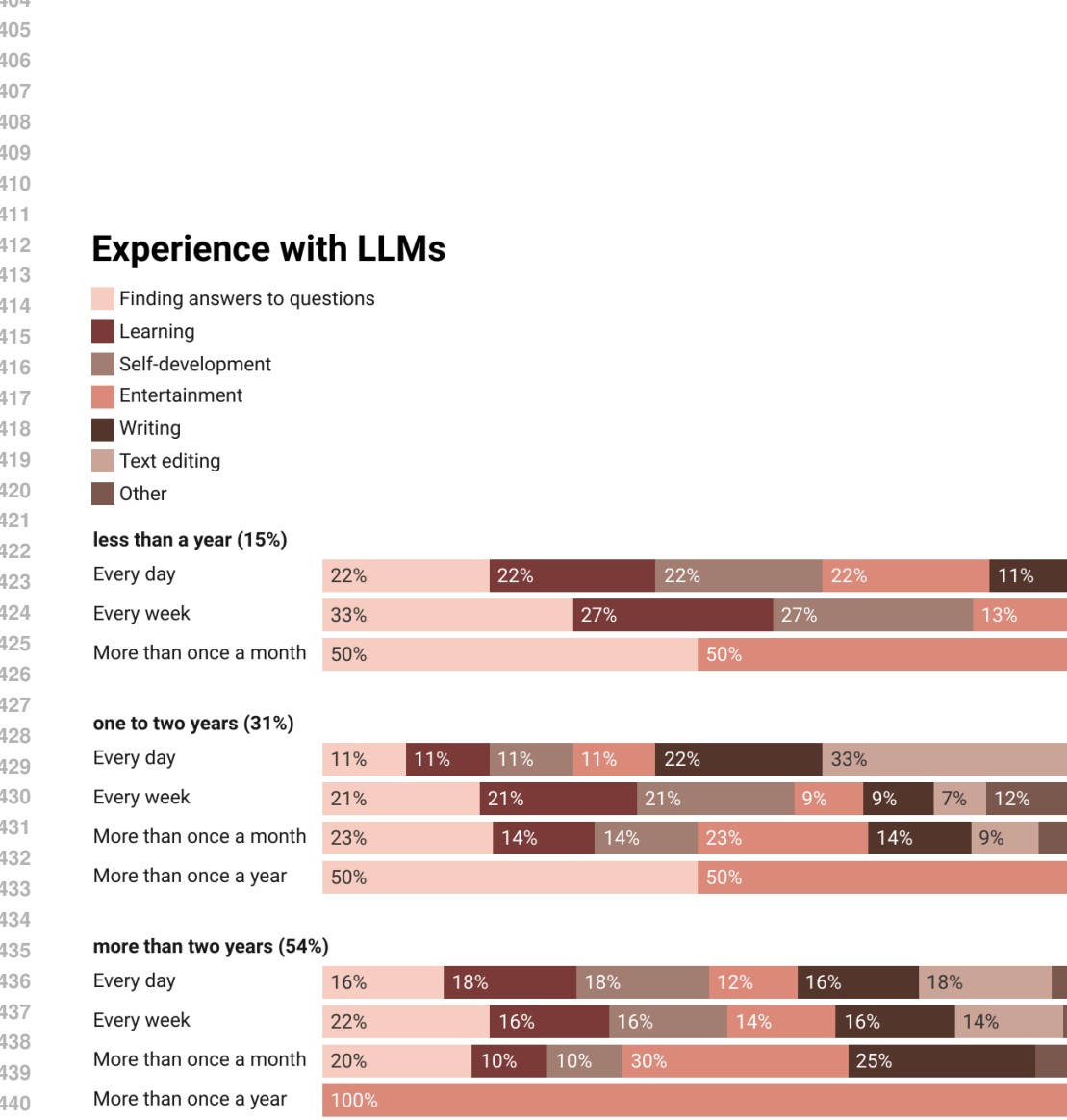

Figure 11: Survey participant distribution by experience with LLMs. The data reveals a community predominantly comprised of experienced LLM users, with the majority having integrated these technologies into their workflows for significant periods. This distribution suggests that the benchmark results largely reflect insights from practitioners with substantial practical knowledge rather than newcomers, lending credibility to the evaluations and observations presented in this study.

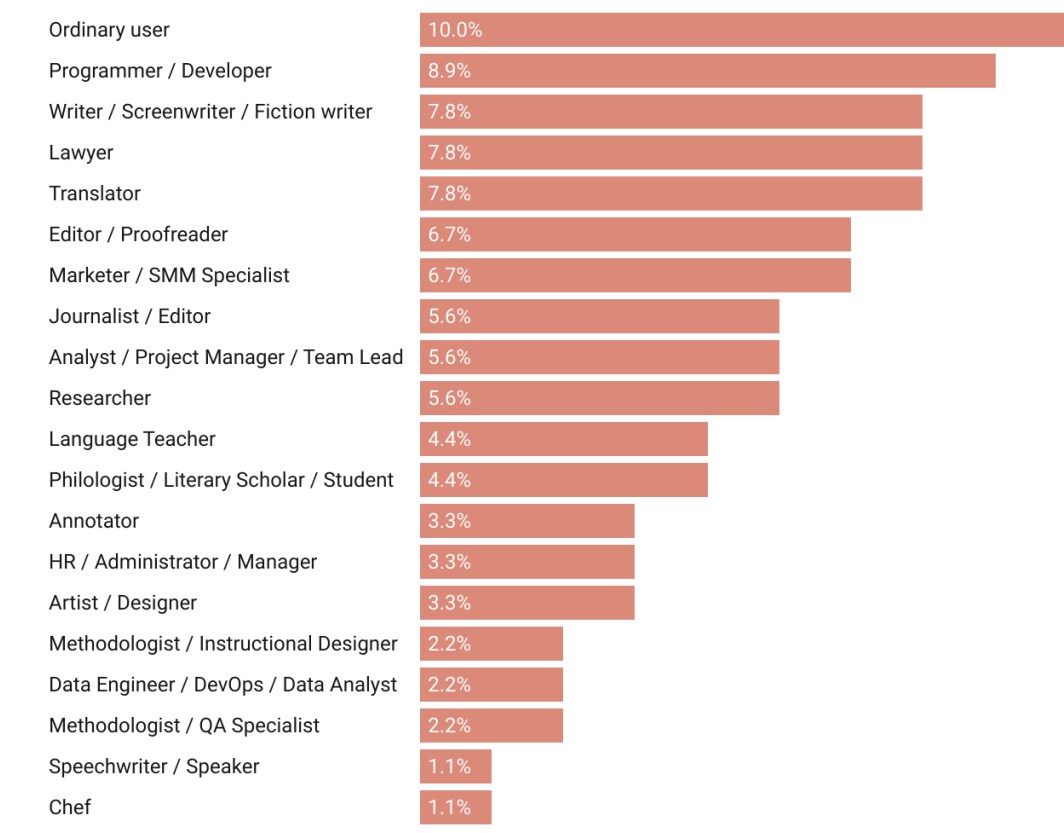

**Professions that would benefit from LLMs**

| Profession | Percentage |
|---|---|
| Ordinary user | 10.0% |
| Programmer / Developer | 8.9% |
| Writer / Screenwriter / Fiction writer | 7.8% |
| Lawyer | 7.8% |
| Translator | 7.8% |
| Editor / Proofreader | 6.7% |
| Marketer / SMM Specialist | 6.7% |
| Journalist / Editor | 5.6% |
| Analyst / Project Manager / Team Lead | 5.6% |
| Researcher | 5.6% |
| Language Teacher | 4.4% |
| Philologist / Literary Scholar / Student | 4.4% |
| Annotator | 3.3% |
| HR / Administrator / Manager | 3.3% |
| Artist / Designer | 3.3% |
| Methodologist / Instructional Designer | 2.2% |
| Data Engineer / DevOps / Data Analyst | 2.2% |
| Methodologist / QA Specialist | 2.2% |
| Speechwriter / Speaker | 1.1% |
| Chef | 1.1% |

*100 people participated in the survey. The participants answered the question: "List the professions that would benefit from LLMs". Participants were free to choose 2 or more options.*

Figure 12: Professional spheres where LLMs may help. This distribution suggests generative AI's greatest value may lie in augmenting knowledge work requiring both structured information processing and creative adaptation. The prominence of ordinary users atop this hierarchy underscores these technologies' democratizing potential. These findings point to areas where focused development efforts and specialized evaluation benchmarks may yield particularly high-value applications.

