# OpenReview forum: "Eye of Judgement: Dissecting the Evaluation  of Russian-speaking LLMs with POLLUX"
_ICLR.cc/2026/Conference — Submitted to ICLR 2026_

### Official Review · Reviewer_6M6V · 2025-10-16

**Soundness:** 2
**Presentation:** 2
**Contribution:** 2
**Rating:** 4
**Confidence:** 2

**Summary:**

The paper introduces POLLUX, an open-source framework for Russian-speaking LLM evaluation. The framework provides 35 task categories along with 22 criteria, and 2115 manually-designed prompts with expert annotations. Finally, 2 LLMs of size 7B and 32B are released to provide automated assessment.

**Strengths:**

The work contributes to Russian LLM evaluation by crafting a high-quality manually crafted dataset, covering a wide-range of tasks and detailed evaluation criteria.

**Weaknesses:**

1. The released models (7B and 32B) do not have high correlations compared with baselines in Table 1. In addition, on many categories the correlation is only around 0.1, so the LLM-as-a-judge score in Table 5 & 6 is not reliable.
2. The benchmark contains many task categories (35), but only around 2K prompts. This can lead to high evaluation variance on some categories with very few prompts.

**Questions:**

How does POLLUX 7B and 32B compare with open-weight models of similar size?

---

> ### Author Response · Authors · 2025-11-26
> **Official Comment by Authors**
>
> We thank the reviewer for the time and effort invested in reading and assessing our work.
>
> >> The released models (7B and 32B) do not have high correlations compared with baselines in Table 1. In addition, on many categories the correlation is only around 0.1, so the LLM-as-a-judge score in Table 5 & 6 is not reliable.
>
> Evaluation on some of the criteria in Table 1 is genuinely challenging, and to assess our LLM-as-a-Judge we compare its results against state-of-the-art models, keeping in mind that for some criteria all models still perform poorly. Furthermore, due to the properties of Spearman’s rank correlation, it is not possible to directly compare the results of Table 1 with those of Overall and Table 3, since they report an overall correlation that takes into account differences between criteria – this may be a special case of Simpson’s paradox. Therefore, for each individual criterion, the correlation will be lower. At the same time, in Tables 5–6 we aggregate scores across multiple criteria and tasks, and Table 3 provides a better assessment of quality under such grouping.
>
> >> How does POLLUX 7B and 32B compare with open-weight models of similar size?
>
> While a comparative analysis against a multitude of smaller, similarly-sized models is feasible, our evaluation strategy deliberately prioritizes benchmarking against larger, state-of-the-art (SOTA) models. This approach is methodologically grounded in demonstrating absolute performance and practical utility, rather than merely establishing relative superiority within a constrained parameter class. By demonstrating that POLLUX, at 7B and 32B parameters, remains competitive with or even challenges models several times its size, we provide more compelling evidence for its architectural efficiency and training data quality.
>
> >>The benchmark contains many task categories (35), but only around 2K prompts. This can lead to high evaluation variance on some categories with very few prompts.
>
> We appreciate the concern regarding potential evaluation variance caused by small category sizes. In our benchmark, however, all categories contain an equal number of prompts. Therefore, a single misclassified example affects each category by the same magnitude, and no category has disproportionally high or low influence on the aggregated score.
> This balanced structure substantially mitigates evaluation variance and ensures fair model comparison across categories.

---

### Official Review · Reviewer_iLCZ · 2025-10-31

**Soundness:** 2
**Presentation:** 1
**Contribution:** 1
**Rating:** 2
**Confidence:** 5

**Summary:**

The paper introduces a new benchmark for evaluating LLM capabilities in the Russian language, using tasks derived from the Russian-speaking portion of WildChat-1M. The authors have performed a large annotation effort, and trained LLM-as-judges to try to mimic expert ratings in an automatic metric setting. The authors report on current LLM performance on the benchmark.

**Strengths:**

The strongest contribution of this paper is a new measure of performance specifically for Russian speaking capabilities, which they use to measure Russian language capabilities of several current models. The measure is the result of a large-scale annotation effort.

**Weaknesses:**

The paper does not make a convincing case as to why it makes a significant contribution to the field of AI.
While the authors frame the benchmark as helping to address the general challenge of evaluating generative tasks (inappropriately, e.g. in the first 2 sentences of the abstract), the paper does not make a significant contribution to new methodology there---which would require deeply surveying existing methodology and demonstrating why any new methodology improves evaluation. However, the paper only superficially covers related work on evaluation methodology.
Even in the domain of Russian language, I would expect a crisp characterization of what are the limitations of current measures of Russian speaking capabilities, and how does this benchmark address those limitations---but again, the related work was quite weak there.
I am also suspicious of the quality of the LLM-as-judge models, which say reference-free is a benefit; however, I would expect that item-specific criteria (e.g., a rubric) would help to align answers rather than relying fully on parametric knowledge.

**Questions:**

How would the authors characterize the limitations of current work on evaluation methodology generally and evaluation artifacts for Russian-speaking capabilities, and how does this work address those limitations?
Can the authors justify the quality of the LLM-as-judge models given their reference-free design?

---

> ### Author Response · Authors · 2025-11-26
> **Official Comment by Authors**
>
> We thank the reviewer for the time and effort invested in reading and assessing our work.
>
> First, we would like to clarify that the abstract, introduction, and conclusion explicitly articulate the contribution of our benchmark. We emphasize that current evaluation practices for open-ended text generation (such as those used in LLM Arena-style comparisons or pairwise SBS setups) do not provide either developers or end-users with a clear understanding of why explicitly a model performs well or poorly on open-ended tasks. Existing methodologies offer coarse, often opaque assessments that fail to reveal model-specific strengths and weaknesses. Our work addresses this gap by offering a fine-grained, expert-defined evaluative framework that is task-agnostic and applicable across languages, including but not limited to Russian. Because the tasks themselves are general generative tasks, the methodology and criteria transfer naturally and provide the community with interpretable, expert-grounded signals that current evaluation pipelines lack.
>
> With respect to Russian-language evaluation specifically, we agree that the body of related work is limited. This is precisely why we highlight the lack of robust, modern benchmarks for open-ended generation in Russian. At the moment, REPA is essentially the only available benchmark in this space, yet it covers a considerably narrower skill set, does not include explicit criteria or rubrics, and relies on error categories that are increasingly outdated given the rapid capabilities growth of contemporary LLMs. In contrast, POLLUX is designed as a next-generation benchmark: it incorporates expert-formulated criteria, offers comprehensive coverage of diverse generative skills, and includes an auto-judge, for which no comparable alternative currently exists in Russian-language evaluation.
>
> Regarding the reviewer’s question about the limitations of existing evaluation methodology more broadly: due to strict page limits, we were not able to provide an extensive standalone survey. We acknowledge that a deeper discussion is valuable, and we plan to extend this section in the camera-ready version. Nonetheless, even within the available space, we outline the key methodological limitations we target in our work: lack of interpretability, insufficient granularity, weak alignment between evaluation signals and expert standards, and scarce resources for non-English languages.
>
> Concerning LLM-as-judge models and their reference-free design: our auto-judge does support the use of rubrics (both the standardized criteria introduced in our methodology and custom rubrics provided by the user.) The ability to incorporate item-specific criteria ensures alignment and reduces reliance on purely parametric knowledge when the task calls for stricter structure. The reference-free mode is an optional benefit, not a constraint; it enables broader applicability while still allowing rubric-conditioned scoring when appropriate. We will clarify this in the revised manuscript.

---

### Official Review · Reviewer_UbPD · 2025-11-01

**Soundness:** 2
**Presentation:** 3
**Contribution:** 3
**Rating:** 4
**Confidence:** 3

**Summary:**

This paper proposes POLLUX, a Russian-based framework for LLM-as-a-judge evaluation with more clearly defined judging criteria. The paper evaluates POLLUX against typical LLM-as-a-judge approaches, in contrast to human expert evaluation.

**Strengths:**

POLLUX focuses on evaluation research for the Russian language, which is less commonly studied compared to English. The proposed method is interesting and demonstrates greater consistency, as it provides explicit judging criteria rather than allowing the model to rank outputs without guidance. This paper contrasts its findings with human expert judgments and evaluates several LLM-as-a-judge baselines. In addition, the provided benchmark resources, including prompts across various difficulty levels and domains, are highly valuable.

**Weaknesses:**

My main concern lies in the gap within the comparison. The models used for POLLUX differ from those used as the baseline LLM-as-a-judge. Therefore, it is unclear whether the differences in results are caused by the framework itself or by the differences in the underlying LLMs. I suggest including an experimental setting where the same models are used, varying only one factor at a time, to ensure a fair comparison.

The benchmark is expert-created. But it is unclear on what categorizes as expert. Who are they and what are the requirement/demography to be considered as 'expert'. Furthermore, no details on how the expert/annotators were hired or paid.

**Questions:**

-

---

> ### Author Response · Authors · 2025-11-26
> **Official Comment by Authors**
>
> We are grateful to the reviewer for the time and careful consideration given to our work!
>
> >>My main concern lies in the gap within the comparison. The models used for POLLUX differ from those used as the baseline LLM-as-a-judge. Therefore, it is unclear whether the differences in results are caused by the framework itself or by the differences in the underlying LLMs. I suggest including an experimental setting where the same models are used, varying only one factor at a time, to ensure a fair comparison.
>
> For POLLUX, we used the T-pro model, which we further fine-tuned on synthetic data. The answers of T-pro are also included in the Zero-Shot Test, so its results cannot be directly compared with those of POLLUX due to self-bias. We demonstrate the existence of this self-bias in Table 6 and show that fine-tuning T-pro according to the POLLUX methodology helps to overcome it. Moreover, POLLUX is trained on the ratings annotated by DeepSeek-V3, which is why in Tables 1-3 we report correlation metrics for both POLLUX and DeepSeek-V3.
>
> >> The benchmark is expert-created. But it is unclear on what categorizes as expert. Who are they and what are the requirement/demography to be considered as 'expert'. Furthermore, no details on how the expert/annotators were hired or paid.
>
> Although comprehensive data on the experts was collected, it was omitted from the main body to avoid an excessively lengthy appendix. We are, however, prepared to provide the full statistical information upon request and to include it in a revised version. A summary of this data is available in Appendix B: Expert Profiles.

---

### Author Response · Authors · 2025-11-27
**Revised version of the paper**

Dear Reviewers,

We sincerely thank you for your valuable time and insightful comments on our work. In response to your feedback, we have thoroughly revised the article.
During the revision period, new data and results became available, which we have incorporated into the current version. The key updates include:

— The addition of new quantitative data and POLLUX judge-models update.
— Enhanced information regarding the expert panel.
— Minor clarifications and elaborations throughout the text to address any ambiguities.

We believe these revisions have significantly strengthened the paper.

---

### Meta-Review · Area_Chair_dhQK · 2026-01-21

**Summary:**

The reviewers raised several concerns across four main areas:

**1. Experimental Design & Fair Comparison**
Reviewer UbPD noted that the comparison between POLLUX and baseline LLM-as-a-judge methods is potentially unfair because different underlying models were used. This makes it unclear whether performance differences stem from the proposed framework or simply from differences in base model capabilities.

**2. Technical Performance & Reliability**
Reviewer 6M6V raised concerns about the low correlation scores of the released judge models (7B and 32B). On many evaluation criteria, correlations are only around 0.1, which calls into question the reliability of the automated scores reported in the main results tables. Additionally, concerns were raised that with 35 task categories but only ~2K prompts, the benchmark may suffer from high variance in categories with few examples.

**3. Contribution Scope & Related Work**
Reviewer iLCZ argued that the paper overclaims its contribution by framing the work as addressing general evaluation challenges, when it primarily provides Russian-specific resources. The related work section was criticized as superficial, lacking both a thorough survey of evaluation methodology and a clear characterization of existing Russian-language evaluation limitations.

**4. Documentation & Transparency**
Multiple reviewers noted insufficient details about the expert annotators—including their qualifications, demographics, and compensation—though authors addressed this partially in their rebuttal by pointing to Appendix B.

**Reviewer Concerns:**

The authors provided rebuttals addressing several concerns, including clarifications about self-bias in comparisons, the availability of expert profiles, and the balanced category structure mitigating variance. However, the core concerns about correlation reliability and contribution scope remain partially unresolved.

**Reviewer Scores:**

**Reviewer UbPD (Original: 4, likely to increase to 5)**
- Fully addressed: Expert qualification transparency (committed to adding comprehensive details in Appendix B)
- Partially addressed: Experimental design confound (explained self-bias rationale but no controlled experiment provided)
- Likelihood: Moderate-to-high chance of score increase given reviewer's initial openness ("would not mind if paper is accepted") and satisfactory resolution of transparency concern

**Reviewer iLCZ (Original: 2, likely unchanged)**
- Fully addressed: Reference-free design misunderstanding (clarified rubric support)
- Not addressed: Fundamental contribution framing disagreement remains unresolved; Literature review weakness (acknowledged, committed to camera-ready improvement)
While the authors clarified that the LLM-as-judge supports rubrics (not purely reference-free), the core concerns about overclaimed contribution scope and superficial related work were not substantively addressed. The authors acknowledged the related work limitation but deferred fixes to camera-ready, which doesn't resolve the current submission's weakness.

**Reviewer 6M6V (Original: 4, likely to increase)**
- Partially addressed: Low correlation reliability (The explanation provides context but doesn't resolve the fundamental reliability concern about low correlations); regarding the request of comparison with similar-sized models, the author justified their choice and did not provide the requested comparison.
Combined with this reviewer's low confidence (2/5) and stated openness to acceptance, these explanations would likely satisfy their concerns.

---

### Decision · Program_Chairs · 2026-01-26

Reject